# TFTF: Training-Free Targeted Flow for Conditional Sampling

**Qianqian Qu** [1]   **Jun S. Liu** [2,3]

## Abstract

We propose a training-free conditional sampling method for flow matching models based on importance sampling. Because a naïve application of importance sampling suffers from weight degeneracy in high-dimensional settings, we modify and incorporate a resampling technique in sequential Monte Carlo (SMC) during intermediate stages of the generation process. To encourage the generated samples to diverge along distinct trajectories, we derive a stochastic flow with adjustable noise strength to replace the deterministic flow at the intermediate stage. Our framework requires no additional training, while providing theoretical guarantees of asymptotic accuracy. Experimentally, our method significantly outperforms existing approaches on conditional sampling tasks for MNIST and CIFAR-10. We further demonstrate the applicability of our approach in higher-dimensional, multimodal settings through text-to-image generation experiments on CelebA-HQ. Code is available at https://github.com/Qqq-22-Rere/tftf.

## 1. Introduction

Flow matching models (Lipman et al., 2022; Liu et al., 2022a) have emerged as powerful generative frameworks capable of learning complex data distributions. When pre-trained on large-scale datasets, these models can generate new samples that (approximately) follow the underlying distribution $p_{\text{data}}(\cdot)$. However, practical applications often require conditional sampling, that is, generating samples that satisfy specific properties. In such cases, the objective shifts to sampling from the conditional distribution $p_{\text{data}}(\cdot|y)$, where $y$ denotes the target property. By Bayes' theorem, this distribution is proportional to $p_{\text{data}}(\cdot)\,\tilde{p}(y|\cdot)$, where $\tilde{p}(y|\cdot)$ denotes the likelihood evaluated at the given conditioning value $y$.

Recent advances in conditional generation can be broadly categorized into two paradigms: training-based methods (Zheng et al., 2023; Holzschuh & Thuerey, 2024) and training-free methods (Ben-Hamu et al., 2024; Wang et al., 2024; Patel et al., 2025; Song et al., 2025). Training-based methods have demonstrated considerable empirical success, yet they require direct access to labeled data for retraining or fine-tuning, and any new conditioning type necessitates re-annotation and additional training. Training-free methods constitute an appealing alternative: rather than modifying the generative model itself, they assume access only to a pretrained unconditional model and a time-independent likelihood function. Fortunately, for a vast majority of conditioning types, high-performing likelihood models are readily available off-the-shelf, enabling seamless incorporation of new conditioning signals without any labeled data. Despite this appeal, existing training-free approaches suffer from several critical limitations: they require differentiation through the ODE solver, exhibit sensitivity to random seed initialization, and lack theoretical guarantees for sampling from the target distribution $p_{\text{data}}(\cdot \mid y)$.

In this paper, we propose a training-free conditional sampling method for flow matching models based on importance sampling (IS) (Liu, 2001). The key idea is to construct a proposal distribution by directly combining a pretrained unconditional model with an off-the-shelf likelihood function $\tilde{p}(y|\cdot)$, then apply importance weighting to correct for the discrepancy from the target $p_{\text{data}}(\cdot|y)$. However, a naïve application of this approach presents two main challenges. First, IS suffers from weight degeneracy in high-dimensional settings (Tokdar & Kass, 2010). Specifically, when the dimensionality of the data space $\mathbb{R}^d$ is large, the majority of the generated samples receive disproportionately low weights and thus contribute negligibly to approximating the target distribution. This issue suggests a potential improvement: evaluating the "quality" of partially generated samples at intermediate stages and terminating the simula-

---

[1]Zhili College, Tsinghua University, Beijing, China [2]Department of Statistics and Data Science, Tsinghua University, Beijing, China [3]Department of Statistics, Harvard University, Cambridge, MA, USA. Correspondence to: Jun S. Liu <junsliu@mail.tsinghua.edu.cn>.

*Proceedings of the 43rd International Conference on Machine Learning*, Seoul, South Korea. PMLR 306, 2026. Copyright 2026 by the author(s).

tion of low-quality ones before completion. Since this early stopping mechanism reduces the total number of samples generated, a possible remedy is to use resampling techniques broadly adopted in Sequential Monte Carlo (SMC) (Liu & Chen, 1995; 1998) to prune away unpromising partial samples and enrich the particle set with promising ones to maintain the desired sample count. This solution, however, introduces a second challenge: the generation process of flow matching is based on ordinary differential equations (ODEs) and follows a fully deterministic dynamics. Consequently, when a partially generated sample is resampled multiple times, the resulting copies will follow the same trajectory and produce identical outputs, failing to explore diverse regions of the target distribution.

To allow multiply sampled particles to traverse distinct generative paths in the future, we derive a stochastic flow with adjustable noise strength to replace the deterministic flow at the intermediate stage. Furthermore, at the final time step, we introduce an extended target distribution that admits $p_{\text{data}}(\cdot|y)$ as its marginal, hence enabling principled reweighting of the generated samples. Through this framework, we obtain a set of properly weighted samples that converge to the target distribution $p_{\text{data}}(\cdot|y)$ with weight balance.

Our approach exhibits several desirable properties. First, it seamlessly integrates the pretrained unconditional model with the time-independent likelihood function $\tilde{p}(y|\cdot)$ in a plug-and-play manner, requiring neither fine-tuning nor retraining. Second, the method is theoretically grounded in IS, enabling the generation of weighted samples that provide an asymptotically exact representation of the target distribution $p_{\text{data}}(\cdot|y)$. Third, by incorporating the SMC resampling technique at intermediate stages, we mitigate the weight degeneracy of IS in high-dimensional settings, thereby ensuring that the generated samples maintain consistently high quality with high effective sample size (ESS) (Liu, 2001). Experimentally, we begin with a toy example to demonstrate that our method produces an accurate particle-based representation of $p_{\text{data}}(\cdot|y)$ in the asymptotic regime. We then evaluate class-conditional sampling on MNIST and CIFAR-10, in which our method generates high-fidelity samples that adhere to specified class labels. Finally, we extend to text-to-image generation on CelebA-HQ, demonstrating the applicability of our approach to higher-dimensional, multimodal settings.

## 2. Background

### 2.1. Flow Matching Models

Suppose a dataset comprises i.i.d. samples drawn from an unknown distribution $p_{\text{data}}(\cdot)$. The objective of generative modeling is to leverage this dataset to learn a model capable

of generating new samples from $p_{\text{data}}(\cdot)$.

Consider the stochastic process constructed as $\{\tilde{X}(t) = t\tilde{X}(1) + (1-t)\tilde{X}(0) : t \in [0,1]\}$, where $\tilde{X}(0) \sim \mathcal{N}(0,I)$ and $\tilde{X}(1) \sim p_{\text{data}}$ are independent. This process induces a continuous-time family of marginal densities $\{f(\cdot,t) : t \in [0,1]\}$ that interpolates between $\mathcal{N}(0,I)$ at $t = 0$ and $p_{\text{data}}(\cdot)$ at $t = 1$. Specifically, for $t \in (0,1)$, this interpolation path is defined through the conditional distribution

$$p(\tilde{x}(t)|\tilde{x}(1)) = \mathcal{N}(\tilde{x}(t); t\tilde{x}(1), (1-t)^2 I), \quad (1)$$

from which the marginal density follows as

$$f(\tilde{x}(t), t) = \int p(\tilde{x}(t)|\tilde{x}(1)) p_{\text{data}}(\tilde{x}(1)) d\tilde{x}(1). \quad (2)$$

Liu et al. (2022a) propose to learn a velocity field $v_\theta(x, t) : \mathbb{R}^d \times [0,1] \to \mathbb{R}^d$ by minimizing the following regression objective:

$$\theta^* = \arg\min_\theta \int_0^1 \mathbb{E}\left[\|\tilde{X}(1) - \tilde{X}(0) - v_\theta(\tilde{X}(t), t)\|^2\right] dt. \quad (3)$$

With sufficient data and model capacity, the optimal velocity field $v_{\theta^*}(x, t)$ recovers the conditional expectation $\mathbb{E}[\tilde{X}(1) - \tilde{X}(0)|\tilde{X}(t) = x]$ almost everywhere.

Generating samples from $p_{\text{data}}(\cdot)$ is done by drawing $x(0)$ from $\mathcal{N}(0, I)$ and integrating the learned velocity field forward in time using the ODE:

$$dx(t) = v_{\theta^*}(x(t), t) dt. \quad (4)$$

The continuity equation (Øksendal, 2003) ensures that the ODE $dX(t) = v_{\theta^*}(X(t), t) dt$ evolves an initial random variable $X(0) \sim \mathcal{N}(0, I)$ forward in time such that the evolving marginal density of $X(t)$ coincides with $f(\cdot, t)$ for all $t \in [0, 1]$. Consequently, up to numerical integration error, the final state $x(1)$ constitutes an exact sample from the data distribution $p_{\text{data}}(\cdot)$.

### 2.2. Density Function along an ODE Path

Let $\{X(t) : t \in [t_1, t_2]\}$ denote a stochastic process governed by the ODE:

$$dX(t) = v(X(t), t) dt, \quad X(t_1) \sim p(\cdot, t_1). \quad (5)$$

We denote by $p(\cdot, t)$ the marginal density function of $X(t)$, which is jointly determined by the velocity field $v(x, t)$ and the initial distribution $p(\cdot, t_1)$.

Given a tractable initial distribution $p(\cdot, t_1)$, the instantaneous change of variables formula (Chen et al., 2018) enables exact computation of the probability density $p(x(t_2), t_2)$ through integration of the following coupled

differential equations:

$$\frac{dx(t)}{dt} = v(x(t), t), \tag{6a}$$

$$\frac{d \log p(x(t), t)}{dt} = -\nabla \cdot v(x(t), t), \tag{6b}$$

where $\nabla \cdot v(x(t), t) = \text{Tr}\left(\frac{\partial v(x(t), t)}{\partial x(t)}\right)$. This yields

$$p(x(t_2), t_2) = p(x(t_1), t_1)e^{-\int_{t_1}^{t_2} \nabla \cdot v(x(t), t) dt}. \tag{7}$$

### 2.3. Sequential Monte Carlo

SMC (Liu & Chen, 1998; Doucet et al., 2001) is a computational framework for approximating a sequence of evolving probability distributions $\{\pi_{t_n}(x_{t_0:t_n})\}_{n=0,1,\dots}$ through sequential simulation of weighted particle sets. This framework comprises two fundamental components: (1) a sequence of target distributions $\{\pi_{t_n}(x_{t_0:t_n})\}_{n=0,1,\dots}$, representing the evolving probability distributions to be approximated, which may be unnormalized; and (2) a sequence of proposal distributions, consisting of an initial distribution $q_{t_0}(x_{t_0})$ and subsequent proposal kernels $\{q_{t_n}(x_{t_n}|x_{t_0:t_{n-1}})\}_{n=1,2,\dots}$.

The algorithm proceeds as follows. At the initial step $t = t_0$, $K$ particles $\{x_{t_0}^{(k)}\}_{k=1}^{K}$ are drawn from $q_{t_0}(\cdot)$ and assigned weights $w_{t_0}^{(k)} = \pi_{t_0}(x_{t_0}^{(k)})/q_{t_0}(x_{t_0}^{(k)})$. For subsequent steps $n = 1, 2, \dots$, the following operations are applied recursively:

- **Resample:** Resample from $\{x_{t_0:t_{n-1}}^{(k)}\}_{k=1}^{K}$ with probabilities proportional to $\{w_{t_{n-1}}^{(k)}\}_{k=1}^{K}$ to produce an equally weighted sample set $\{x_{t_0:t_{n-1}}^{(*k)}\}_{k=1}^{K}$.

- **Propagate:** For each $k$, draw $x_{t_n}^{(k)}$ from the proposal $q_{t_n}(\cdot|x_{t_0:t_{n-1}}^{(*k)})$ and append it to $x_{t_0:t_{n-1}}^{(*k)}$ to form $x_{t_0:t_n}^{(k)} = (x_{t_0:t_{n-1}}^{(*k)}, x_{t_n}^{(k)})$.

- **Reweight:** Compute the importance weight $w_{t_n}^{(k)} = \pi_{t_n}(x_{t_0:t_n}^{(k)})/[\pi_{t_{n-1}}(x_{t_0:t_{n-1}}^{(*k)}) \cdot q_{t_n}(x_{t_n}^{(k)}|x_{t_0:t_{n-1}}^{(*k)})]$ to produce the weighted sample set $\{(x_{t_0:t_n}^{(k)}, w_{t_n}^{(k)})\}_{k=1}^{K}$ for time $t_n$.

Through this iterative process, SMC propagates the particle approximation forward in time: at each step $t = t_n$, particles from the previous time $t_{n-1}$ are resampled according to their weights, extended via the proposal distribution, and then reweighted to align with the current target distribution $\pi_{t_n}(\cdot)$. For each time $t_n$, the resulting particle set $\{(x_{t_0:t_n}^{(k)}, w_{t_n}^{(k)})\}_{k=1}^{K}$ provides a properly weighted sample-based approximation that converges to the target distribution $\pi_{t_n}(\cdot)$ as $K \to \infty$ under appropriate regularity conditions (Del Moral, 2004; Chopin et al., 2020).

*Remark* 2.1. SMC encompasses a broader family of variants and techniques than the approach presented here; interested readers are referred to Liu & Chen (1998) for further exposition.

## 3. Training-Free Conditional Sampling

Suppose we have a pretrained velocity field $v_{\theta*}$ that enables sampling from $p_{\text{data}}(\cdot)$. The objective of training-free conditional generation is to sample from the conditional distribution $p_{\text{data}}(\cdot|y) \propto p_{\text{data}}(\cdot)\tilde{p}(y|\cdot)$ by leveraging an off-the-shelf model $\tilde{p}(y|\cdot)$ (e.g., a time-independent classifier trained on noise-free images), without requiring any additional training.

Throughout this section, we denote by $\{\tilde{X}(t) : t \in [0, 1]\}$ the stochastic process satisfying the following conditions: (i) $\tilde{X}(1) \sim p_{\text{data}}$ and $\tilde{X}(0) \sim \mathcal{N}(0, I)$ are mutually independent; (ii) intermediate states are constructed via linear interpolation, that is, $\tilde{X}(t) = t\tilde{X}(1) + (1 - t)\tilde{X}(0)$; and (iii) conditioned on $\tilde{X}(1) = \tilde{x}(1)$, the variable $Y$ satisfies $p_{Y|\tilde{X}(1)}(y|\tilde{x}(1)) = \tilde{p}(y|\tilde{x}(1))$ and is conditionally independent of $\tilde{X}(t)$ for all $t \in [0, 1)$. A summary of the notation used throughout this paper is also provided in Table 4.

### 3.1. Importance Sampling-Based Conditional Generation

#### 3.1.1. CONSTRUCTING CONDITIONAL VELOCITY FIELD

In the ideal scenario, the ODE that progressively transforms the noise distribution $\mathcal{N}(0, I)$ at $t = 0$ into the target distribution $p_{\text{data}}(\cdot|y)$ at $t = 1$ takes the form $dX(t) = v_c^*(X(t), t)dt$, where the optimal conditional velocity field $v_c^*(x(t), t)$ is defined as $\mathbb{E}[\tilde{X}(1) - \tilde{X}(0)|\tilde{X}(t) = x(t), Y = y]$ (see Section A.1 for the proof). To perform training-free conditional sampling building upon the pretrained unconditional model $v_{\theta*}$, we decompose the optimal conditional velocity field $v_c^*$ as stated in the following proposition.

**Proposition 3.1** (Proof in Section A.2). *For $t \in (0, 1]$, the optimal conditional velocity field $v_c^*(x(t), t)$ can be decomposed as*[1]

$$v_c^*(x(t), t) = v_{\theta*}(x(t), t) + \frac{1 - t}{t} \cdot \nabla_{x(t)} \log p_{Y|\tilde{X}(t)}(y|x(t)). \tag{8}$$

However, computing the term $p_{Y|\tilde{X}(t)}(y|x(t))$ requires direct evaluation of the intractable dependency between the conditioning variable $Y$ and the intermediate state $\tilde{X}(t)$. To address this issue, we propose to project the noise-corrupted intermediate state $x(t)$ to a predicted final state $\hat{x}_1(x(t))$ via

---

[1]At the boundary, $v_c^*(x, 0) = \lim_{t \to 0^+} v_c^*(x, t)$.

a single Euler step[2]:

$$\hat{x}_1(x(t)) := x(t) + (1 - t) \cdot v_{\theta*}(x(t), t), \qquad (9)$$

and replace $p_{Y|\tilde{X}(t)}(y|x(t))$ with $p_{Y|\tilde{X}(1)}(y|\hat{x}_1(x(t))) = \tilde{p}(y|\hat{x}_1(x(t)))$. This yields the practical conditional velocity field

$$v_c(x(t), t) := v_{\theta*}(x(t), t) +$$
$$\frac{1 - t}{t} \cdot \beta(t) \cdot \nabla_{x(t)} \log \tilde{p}(y|\hat{x}_1(x(t))), \qquad (10)$$

where $\beta(t)$ represents a user-defined guidance scale. Intuitively, the pretrained velocity term $v_{\theta*}(x(t), t)$ of Equation (10) directs particles toward regions of natural-like data (i.e., the manifold underlying $p_{\text{data}}(\cdot)$), while the gradient term further steers particles toward subregions that satisfy the desired conditioning property $y$.

### 3.1.2. IMPORTANCE WEIGHTING AND WEIGHT DEGENERACY

Let $p_1(\cdot)$ denote the terminal distribution at $t = 1$ obtained by solving the ODE $dX(t) = v_c(X(t), t)dt$ with initial distribution $\mathcal{N}(0, I)$ at $t = 0$. Due to the gap between $v_c$ (10) and $v_c^*$ (8), the resulting distribution $p_1(\cdot)$ does not perfectly match the target distribution $p_{\text{data}}(\cdot|y)$.

To correct for this discrepancy, IS (Liu, 2001) can be applied to reweight the generated samples. Concretely, let $\{x(0)^{(k)}\}_{k=1}^K$ be initial states drawn from $\mathcal{N}(0, I)$, and let $\{x(1)^{(k)}\}_{k=1}^K$ denote the corresponding terminal states obtained by solving $dx(t) = v_c(x(t), t)dt$ from $t = 0$ to $t = 1$. Each terminal sample is then assigned a weight

$$w^{(k)} = \frac{p_{\text{data}}(x(1)^{(k)}|y)}{p_1(x(1)^{(k)})} \propto \frac{p_{\text{data}}(x(1)^{(k)}) \cdot \tilde{p}(y|x(1)^{(k)})}{p_1(x(1)^{(k)})}.$$

Here, $p_{\text{data}}(\cdot)$ in the numerator and $p_1(\cdot)$ in the denominator can both be computed via the instantaneous change of variables formula (Chen et al., 2018) presented in Section 2.2. The resulting samples $\{(x(1)^{(k)}, w^{(k)})\}_{k=1}^K$ are then properly weighted with respect to the target distribution $p_{\text{data}}(\cdot|y)$.

However, direct application of this approach is precluded by weight degeneracy (Tokdar & Kass, 2010), which refers to the tendency for importance weights to become highly skewed in high-dimensional settings. In an extreme yet common scenario, a single particle's weight dominates all others, causing the ESS, defined as $(\sum w^{(k)})^2 / \sum (w^{(k)})^2$ (Liu, 2001), to approach unity. This can be interpreted as the $K$ weighted samples being effectively equivalent to a single sample drawn from the target distribution, which is a clearly inefficient use of computational resources.

---

[2]This one-step Euler look-ahead projection admits an alternative interpretation as a conditional expectation. Specifically, it follows readily from Eq. (3) that $\hat{x}_1(x(t)) = \mathbb{E}[\tilde{X}(1) | \tilde{X}(t) = x(t)]$, which is the MMSE-optimal point estimate of the final state.

## 3.2. Training-Free Targeted Flow

To mitigate the aforementioned weight degeneracy problem inherent to high-dimensional IS, we propose to incorporate a resampling technique in SMC (Liu & Chen, 1998) at intermediate stages of the generation process. The central insight is that appropriate reweighting and resampling strategies at intermediate stages enable early termination of unpromising partial samples while enriching the particle set with high-quality candidates. This mechanism concentrates computational resources on propagating promising trajectories, ultimately yielding high-fidelity, condition-compliant samples with well-balanced weights.

**Introducing stochasticity into flow matching.** As introduced in Section 2.1, the generation process in the original flow matching framework is ODE-based and therefore fully deterministic once the initial point is specified. This determinism prevents resampled partial trajectories from branching into varied areas of the target distribution.

To overcome this obstacle, we propose to transform the deterministic flow into a stochastic flow by establishing a connection through the Fokker-Planck-Kolmogorov equation (Särkkä & Solin, 2019). Our approach is inspired by Song et al. (2020), who derived the Probability Flow ODE (PF-ODE) from the forward SDE in score-based diffusion models (Sohl-Dickstein et al., 2015; Song & Ermon, 2019; Ho et al., 2020). In the present work, we pursue the converse direction: we convert the deterministic ODE of flow matching models (Lipman et al., 2022; Liu et al., 2022a) into a family of SDEs, thereby introducing the necessary stochasticity for incorporating resampling.

**Proposition 3.2** (Proof in Section B.1). *Let $\{X_1(t) : t \in [0, 1]\}$ denote the stochastic process defined by the ODE:*

$$dX_1(t) = v_{\theta*}(X_1(t), t)dt, \qquad (11)$$

*where $v_{\theta*}$ is the pretrained velocity field and $X_1(0) \sim \mathcal{N}(0, I)$. This process induces a continuous-time family of marginal probability densities $\{f_t(\cdot) : t \in [0, 1]\}$ satisfying $f_1(\cdot) = p_{data}(\cdot)$. Consider the following SDE:*

$$dX_2(t) = \{\alpha(t)[-X_2(t) + tv_{\theta*}(X_2(t), t)]$$
$$+ v_{\theta*}(X_2(t), t)\} dt + \sqrt{2(1 - t)\alpha(t)} \, dB(t), \qquad (12)$$

*where $B(t)$ denotes standard Brownian motion. Then, given initial condition $X_2(0) \sim \mathcal{N}(0, I)$, this SDE evolves the process forward in time such that the marginal density of $X_2(t)$ coincides with $f_t(\cdot)$ for all $t \in [0, 1]$, and thus matches $p_{data}(\cdot)$ at the terminal time $t = 1$.*

The nonnegative function $\alpha(t)$ in Equation (12) controls the level of stochasticity: setting $\alpha(t) = 0$ recovers the original deterministic flow matching framework, while $\alpha(t) > 0$

introduces stochasticity into the generation process through the Brownian motion term. This stochasticity is essential for performing resampling, as it enables the particles to spread in different directions. Remarkably, a particular specification of $\alpha(t)$ corresponds to Anderson's reverse-time SDE (1982); see Section B.2 for the proof.

**Strategic placement of resampling at intermediate stages.** The generation dynamics exhibit distinct characteristics across different periods, which motivates our design of the sampling procedure:

- **Early period** $[0, t_{n_1}]$: Samples are too noisy to provide reliable information about the final state.

- **Middle period** $[t_{n_1}, t_{n_2}]$: Key features emerge gradually, making this the critical period for conditional generation.

- **Late period** $[t_{n_2}, 1]$: Coarse-scale features have been resolved, with only fine-scale details yet to be refined.

Based on these observations, we implement resampling exclusively over the intermediate interval $[t_{n_1}, t_{n_2}]$, where $0 < t_{n_1} < t_{n_2} < 1$. During the early and late periods, we employ ODE-based simulation in lieu of SDE, since the Brownian motion term becomes superfluous when resampling is not performed. Furthermore, we utilize the pretrained velocity field $v_{\theta^*}$ rather than the conditional velocity field $v_c$ to solve the ODE during the early and late periods, thereby circumventing the computational overhead associated with the gradient computation in Equation (10).

**Designing target and proposal distributions at intermediate stages.** For conditional generation, our goal is to sample from $p_{\text{data}}(\cdot|y) \propto p_{\text{data}}(\cdot)\tilde{p}(y|\cdot)$, where the first factor $p_{\text{data}}(\cdot)$ assigns density to regions consistent with the unconditional distribution (measuring naturalness) and the second factor $\tilde{p}(y|\cdot)$ assigns density to regions satisfying the conditioning constraint $y$ (measuring alignment). In light of this, we design the (unnormalized) intermediate target distribution $\pi_{t_n}(x_{t_{n_1}:t_n})$ to measure both naturalness and alignment of partially generated samples. Specifically, we adopt the following form:

$$f_{t_{n_1}}(x_{t_{n_1}}) \left[ \prod_{i=n_1+1}^{n} h_{t_i}(x_{t_i}|x_{t_{i-1}}) \right] \tilde{p}(y|\hat{x}_1(x_{t_n})), \quad (13)$$

where $f_{t_{n_1}}(\cdot)$ denotes the marginal density at $t = t_{n_1}$ induced by ODE (11) with initial distribution $\mathcal{N}(0, I)$, $h_{t_i}(\cdot|x_{t_{i-1}})$ denotes the conditional distribution of $X_2(t_i)$ given $X_2(t_{i-1}) = x_{t_{i-1}}$ under SDE (12), and $\hat{x}_1(x_{t_n})$ denotes the one-step Euler projection of the intermediate state $x_{t_n}$ to the terminal time $t = 1$:

$$\hat{x}_1(x_{t_n}) = x_{t_n} + (1 - t_n) \cdot v_{\theta^*}(x_{t_n}, t_n). \quad (14)$$

The decomposition in Equation (13) has an intuitive interpretation. The product $f_{t_{n_1}}(x_{t_{n_1}}) \left[ \prod_{i=n_1+1}^{n} h_{t_i}(x_{t_i}|x_{t_{i-1}}) \right]$ measures the naturalness of the trajectory $x_{t_{n_1}:t_n}$, i.e., its consistency with the unconditional generation dynamics prescribed by SDE (12). Meanwhile, the term $\tilde{p}(y|\hat{x}_1(x_{t_n}))$ serves as a look-ahead function that predicts the consistency of the final state with the conditioning constraint $y$ by projecting the current state $x_{t_n}$ forward to completion.

The proposal distributions at intermediate stages emerge naturally from our framework. For $t = t_{n_1}$, we set the proposal distribution $q_{t_{n_1}}(\cdot)$ to $f_{t_{n_1}}(\cdot)$, which is obtained by evolving ODE (11) over $[0, t_{n_1}]$. For subsequent steps where $t_{n_1} < t_n \leq t_{n_2}$, the proposal kernel $q_{t_n}(\cdot|x_{t_{n-1}})$ is given by the conditional distribution of $X_2(t_n)$ given $X_2(t_{n-1}) = x_{t_{n-1}}$ under SDE (12), with $v_{\theta^*}$ replaced by the conditional velocity field $v_c$ constructed in Equation (10). This design ensures that the proposal kernel incorporates the conditioning information $y$ when proposing next states, thereby guiding particles toward regions of high target density.

**The TFTF algorithm and final reweighting.** The preceding analysis yields our algorithm, the Training-Free Targeted Flow (TFTF). As outlined in Algorithm 1, we initialize TFTF by sampling from $\mathcal{N}(0, I)$ and integrating ODE (11) over the interval $[0, t_{n_1}]$, yielding samples from the initial proposal distribution $q_{t_{n_1}}(\cdot) = f_{t_{n_1}}(\cdot)$. The particles are then propagated over the interval $[t_{n_1}, t_{n_2}]$ using the intermediate proposal and target distributions constructed above, with iterative reweighting and resampling. Finally, we revert to the deterministic flow (11) over $[t_{n_2}, 1]$. At the final time step $t = 1$, we define the (unnormalized) final extended target distribution as

$$Z(x_{t_{n_1}:t_{n_2}}) \cdot e^{-\int_{t_{n_2}}^{1} \nabla \cdot v_{\theta^*}(x_t, t)\, dt} \cdot \tilde{p}(y|x_1), \quad (15)$$

where $Z(x_{t_{n_1}:t_{n_2}})$ is given by

$$f_{t_{n_1}}(x_{t_{n_1}}) \left[ \prod_{i=n_1+1}^{n_2} h_{t_i}(x_{t_i}|x_{t_{i-1}}) \right]. \quad (16)$$

This yields the final incremental weight

$$\frac{Z(x_{t_{n_1}:t_{n_2}}) \cdot e^{-\int_{t_{n_2}}^{1} \nabla \cdot v_{\theta^*}(x_t, t)\, dt} \cdot \tilde{p}(y|x_1)}{Z(x_{t_{n_1}:t_{n_2}}) \cdot \tilde{p}(y|\hat{x}_1(x_{t_{n_2}})) \cdot e^{-\int_{t_{n_2}}^{1} \nabla \cdot v_{\theta^*}(x_t, t)\, dt}}$$

$$= \frac{\tilde{p}(y|x_1)}{\tilde{p}(y|\hat{x}_1(x_{t_{n_2}}))}, \quad (17)$$

which corrects for the proposal-target mismatch (see Section C.2 for the complete derivation). Notably, the exponential factors in the numerator and denominator of Equation (17) cancel because we use the pretrained velocity field $v_{\theta^*}$ rather than $v_c$ over $[t_{n_2}, 1]$, thereby avoiding the need for divergence calculation.

Crucially, the final extended target distribution (15) contains $p_{\text{data}}(\cdot|y)$ as its marginal; specifically,

$$\int Z(x_{t_{n_1}:t_{n_2}}) \cdot e^{-\int_{t_{n_2}}^{1} \nabla \cdot v_{\theta^*}(x_t, t)\, dt} \cdot \tilde{p}(y|x_1)\, dx_{t_{n_1}:t_{n_2-1}}$$
$$= p_{\text{data}}(x_1)\tilde{p}(y|x_1), \quad (18)$$

with the full derivation provided in Section C.1. Consequently, Algorithm 1 can approximate $p_{\text{data}}(\cdot|y)$ with asymptotic accuracy, as established in the following proposition.

**Proposition 3.3** (Proof in Section D.1). *For Algorithm 1, suppose that the intermediate weight functions $\{W_{t_n}\}_{n=n_1}^{n_2}$ as well as the final incremental weight $\frac{\tilde{p}(y|X_1)}{\tilde{p}(y|\hat{x}_1(X_{t_{n_2}}))}$ are bounded. Then, for any $\phi \in \mathcal{C}_b(\mathbb{R}^d)$,*

$$\sum_{k=1}^{K} \bar{W}_1^{(k)} \phi(X_1^{(k)}) \xrightarrow{a.s.} \mathbb{E}_{p_{\text{data}}(\cdot|y)}[\phi] \quad as\ K \to \infty. \quad (19)$$

*Remark* 3.4. The implementation details of Algorithm 1, along with verification of the boundedness conditions in Proposition 3.3, are provided in Section D.2.

Given the asymptotic accuracy guarantee of Algorithm 1, approximation quality with respect to the target distribution improves as the number of samples $K$ increases. To scale TFTF to larger sample sizes, it is therefore desirable to integrate distributed sampling results in a principled manner. To this end, we propose in Section E a nested implementation of our method, termed Nested TFTF (Algorithm 2), that retains asymptotic accuracy under finite particle count constraints for each inner-level sampling operation.

To demonstrate the pruning and enrichment effects achieved by incorporating resampling at intermediate stages, we present results for conditional sampling targeting the "automobile" class on the CIFAR-10 dataset in Figure 1. Figure 1a shows the results obtained through direct application of IS while Figure 1b displays the results obtained using Algorithm 1 with $K = 16$. Comparison of the two panels reveals that the incorporation of resampling effectively increases the ESS and enriches the generated sample set with diverse, high-fidelity, and condition-adherent samples.

# 4. Experiments

We first verify the asymptotic accuracy of our method on a toy example (Section 4.1), then evaluate class-conditional sampling on MNIST (LeCun et al., 2002) and CIFAR-10 (Krizhevsky et al., 2009) (Section 4.2), and finally conduct text-to-image generation on CelebA-HQ (Karras et al., 2017) (Section 4.3). Additional experimental details are provided in Section F.

---

**Algorithm 1** Training-Free Targeted Flow (TFTF)

**Require:** Pretrained velocity field $v_{\theta^*}$, likelihood function $\tilde{p}(y|\cdot)$, intermediate proposal distributions $\{q_{t_n}\}_{n=n_1}^{n_2}$, intermediate target distributions $\{\pi_{t_n}\}_{n=n_1}^{n_2}$, number of samples $K$

1: Sample $x_0^{(k)} \sim \mathcal{N}(0, I)$
2: Solve $dx_t = v_{\theta^*}(x_t, t)dt$ from $t = 0$ to $t = t_{n_1}$ to obtain $x_{t_{n_1}}^{(k)}$
3: Compute $w_{t_{n_1}}^{(k)} = \frac{\pi_{t_{n_1}}(x_{t_{n_1}}^{(k)})}{q_{t_{n_1}}(x_{t_{n_1}}^{(k)})}$
4: Normalize $\bar{w}_{t_{n_1}}^{(k)} = \frac{w_{t_{n_1}}^{(k)}}{\sum_{j=1}^{K} w_{t_{n_1}}^{(j)}}$
5: **for** $n = n_1 + 1, \ldots, n_2$ **do**
6:     Resample $\{x_{t_{n_1}:t_{n-1}}^{(k)}\}_{k=1}^{K}$ using $\{\bar{w}_{t_{n-1}}^{(k)}\}_{k=1}^{K}$ to obtain $\{x_{t_{n_1}:t_{n-1}}^{(*k)}\}_{k=1}^{K}$
7:     Sample $x_{t_n}^{(k)} \sim q_{t_n}(\cdot|x_{t_{n-1}}^{(*k)})$
8:     Set $x_{t_{n_1}:t_n}^{(k)} = (x_{t_{n_1}:t_{n-1}}^{(*k)}, x_{t_n}^{(k)})$
9:     Compute $w_{t_n}^{(k)} = \frac{\pi_{t_n}(x_{t_{n_1}:t_n}^{(k)})}{q_{t_n}(x_{t_n}^{(k)}|x_{t_{n-1}}^{(*k)}) \cdot \pi_{t_{n-1}}(x_{t_{n_1}:t_{n-1}}^{(*k)})}$
10:    Normalize $\bar{w}_{t_n}^{(k)} = \frac{w_{t_n}^{(k)}}{\sum_{j=1}^{K} w_{t_n}^{(j)}}$
11: **end for**
12: Solve $dx_t = v_{\theta^*}(x_t, t)dt$ from $t = t_{n_2}$ to $t = 1$ to obtain $x_1^{(k)}$
13: Compute $w_1^{(k)} = \bar{w}_{t_{n_2}}^{(k)} \cdot \frac{\tilde{p}(y|x_1^{(k)})}{\tilde{p}(y|\hat{x}_1(x_{t_{n_2}}^{(k)}))}$
14: Normalize $\bar{w}_1^{(k)} = \frac{w_1^{(k)}}{\sum_{j=1}^{K} w_1^{(j)}}$
15: **return** Weighted samples $\{(x_1^{(k)}, \bar{w}_1^{(k)})\}_{k=1}^{K}$

---

## 4.1. Toy Example

We consider a toy example where $p_{\text{data}}$ is specified as $\frac{3}{4}\mathcal{N}((-\sqrt{2}, -\sqrt{2}), \frac{1}{4}I) + \frac{1}{4}\mathcal{N}((\sqrt{2}, \sqrt{2}), \frac{1}{4}I)$. This specification admits an analytical form for $v_{\theta^*}(x, t) = \mathbb{E}[\tilde{X}(1) - \tilde{X}(0)|\tilde{X}(t) = x]$, which exactly pushes forward $\mathcal{N}(0, I)$ to $p_{\text{data}}$. By employing this closed-form velocity field, we eliminate the impact of neural network approximation errors from unconditional model pre-training on our evaluation.

We define the conditional distribution of $Y$ given $\tilde{X}(1)$ as $\tilde{p}(y|\tilde{x}(1)) := \mathcal{N}(y; \tilde{x}(1), \frac{1}{4}I)$. Given $y = (0, 0)$, we can analytically compute the ground-truth conditional distribution $p_{\text{data}}(\cdot|y)$ as $\frac{3}{4}\mathcal{N}((-\frac{\sqrt{2}}{2}, -\frac{\sqrt{2}}{2}), \frac{1}{8}I) + \frac{1}{4}\mathcal{N}((\frac{\sqrt{2}}{2}, \frac{\sqrt{2}}{2}), \frac{1}{8}I)$. From this distribution, we directly draw 20,000 i.i.d. samples as the gold standard (cf. Figure 2a). For our proposed method (Algorithm 1) and the baseline approaches, we assume access only to $v_{\theta^*}(x, t)$ and $\tilde{p}(y|\cdot)$, and generate 20,000 samples with each method. As illustrated in Figure 2b, the samples generated by our method correctly recover the means, variances, and relative weights between

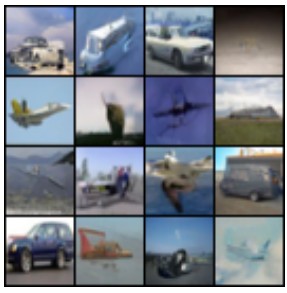 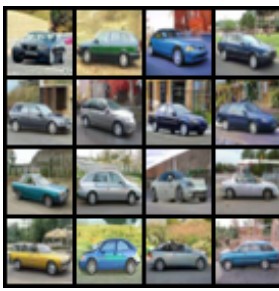

*(a)* Direct IS (ESS: 1.37/16)      *(b)* TFTF (ESS: 16.00/16)

*Figure 1.* Effect of incorporating resampling at intermediate stages of the generation process.

*Table 1.* Class-conditional sampling results on MNIST. $\text{ACC}_L$: likelihood-specifying classifier accuracy; $\text{ACC}_E$: external classifier accuracy; ESS: effective sample size.

| METHOD | $\text{ACC}_L$ | $\text{ACC}_E\uparrow$ | ESS |
|---|---|---|---|
| FMPS | 62.86% | 39.24% | – |
| FLOWCHEF | 91.15% | 23.57% | – |
| TFTF (OURS) | 99.66% | **97.12%** | 15.99/16 |

the two Gaussian components of $p_{\text{data}}(\cdot|y)$. In contrast, the sampling distributions produced by other baseline methods demonstrate varying degrees of bias (cf. Figures 2c to 2e).

## 4.2. Class-Conditional Sampling

For the MNIST dataset, we evaluate the performance of TFTF (Algorithm 1) with $K = 16$. Figure 3a presents samples generated by our algorithm within a single batch, which are consistently authentic and conform to the specified class label, whereas baseline methods exhibit inconsistent generation quality (cf. Figures 3b and 3c).

Because the conditional velocity field $v_c$ (10) combines the pretrained velocity $v_{\theta*}$ with the likelihood gradient, one might worry the samples simply exploit the likelihood-specifying classifier (i.e., the guidance classifier) rather than reflect true class semantics. We therefore assess classification accuracy with an external classifier (denoted $\text{ACC}_E$). As shown in Table 1, our method maintains high accuracy for both the likelihood-specifying classifier (denoted $\text{ACC}_L$) and the external classifier, whereas all baselines show large drops in $\text{ACC}_E$ relative to $\text{ACC}_L$.

We further conduct ablation studies examining the effect of $\alpha(t)$ in SDE (12) and the particle count $K$. For comparison, we also include results from Twisted Diffusion Sampler (TDS) (Wu et al., 2023), another method that generates correlated samples in a single batch, albeit based on the score-based diffusion framework. Figures 4a and 4b present the intra-batch diversity (measured by mean pairwise $\ell_2$ distance) and $\text{ACC}_E$ for $\alpha(t) \in \{1/t, 2/t, 4/t, 8/t\}$. As shown, both metrics generally increase with the scale

*Table 2.* Class-conditional sampling results on CIFAR-10. $\text{ACC}_L$: likelihood-specifying classifier accuracy; $\text{ACC}_E$: external classifier accuracy; IS: Inception Score; $\mathcal{W}_2$: intra-class feature-space Wasserstein-2 distance (averaged across classes).

| METHOD | $\text{ACC}_L$ | $\text{ACC}_E\uparrow$ | IS$\uparrow$ | $\mathcal{W}_2\downarrow$ |
|---|---|---|---|---|
| FMPS | 33.81% | 31.81% | 7.22 ±0.08 | 99.59 |
| FLOWCHEF | 98.04% | 50.13% | 7.54 ±0.05 | 63.69 |
| TFTF (OURS) | 92.56% | **92.82%** | **8.88** ±0.08 | **32.61** |

*Table 3.* Text-to-image generation results on CelebA-HQ. $\text{CLIP}_L$: similarity score measured under the likelihood-specifying CLIP model; $\text{ACC}_E$: average all-attributes accuracy under the external classifier; ESS: effective sample size (averaged across prompts).

| METHOD | $\text{CLIP}_L$ | $\text{ACC}_E\uparrow$ | ESS |
|---|---|---|---|
| FMPS | 0.307 | 68.69% | – |
| FLOWCHEF | 0.305 | 41.50% | – |
| TFTF (OURS) | 0.295 | **80.99%** | 21.77/25 |

of $\alpha(t)$, since larger values enable the particles to diverge more substantially and exhibit stronger exploratory behavior. However, these benefits are not unbounded, as excessively large $\alpha(t)$ values induce unstable intermediate weights and accumulate greater numerical errors during SDE integration. Figure 4c plots $\text{ACC}_E$ against varying particle counts. As illustrated, our method's $\text{ACC}_E$ increases with the particle count, and with a single particle, it surpasses all baseline methods. Samples generated under different $\alpha(t)$ configurations are provided in Figure 8.

For the CIFAR-10 dataset, we assess the ability of our method to approximate the target distribution at higher particle counts, using Nested TFTF (Algorithm 2) with $K = 16$ and $M = 1000$. As reported in Table 2, for a fixed total sample size, our algorithm markedly outperforms the baselines in terms of $\text{ACC}_E$, $IS$, and feature-space Wasserstein-2 distance.

Additionally, Section G.1 presents a simplified procedure for computing $v_c$ in (10) that avoids backpropagation through the velocity-field network. This approach reduces both computation time and GPU memory consumption, thereby enabling the parallel propagation of a larger number of particles, while preserving theoretical convergence guarantees and empirical performance. Section G.1 also provides a detailed analysis of computational complexity across the compared methods (cf. Table 6). Section G.2 further reports sensitivity analysis of the resampling interval $[t_{n_1}, t_{n_2}]$ using TFTF (Algorithm 1) with a finite particle budget.

## 4.3. Text-to-Image Generation

For the CelebA-HQ dataset ($256 \times 256 \times 3$), we conduct text-to-image generation experiments using the accelerated version (cf. Section G.1) of Algorithm 1 with $K = 25$. Given

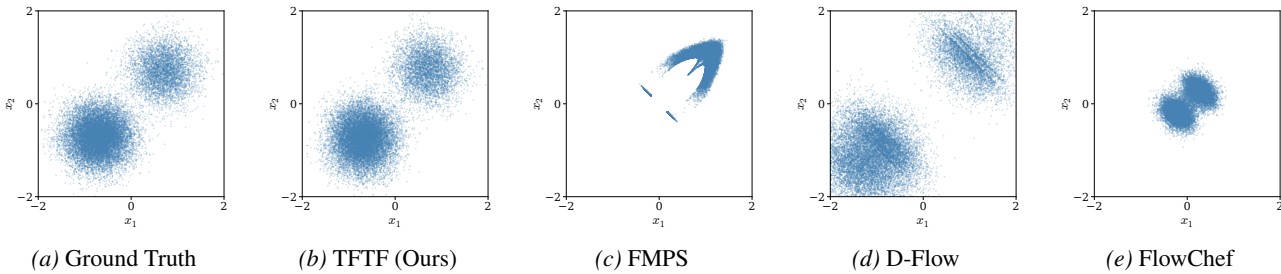

*(a)* Ground Truth     *(b)* TFTF (Ours)     *(c)* FMPS     *(d)* D-Flow     *(e)* FlowChef

*Figure 2.* Samples generated by different methods on the toy example.

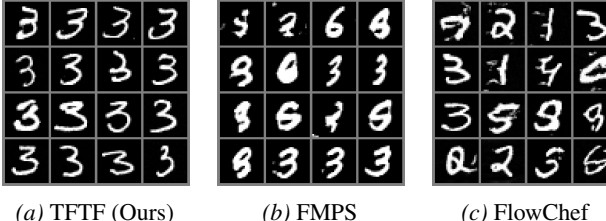

*(a)* TFTF (Ours)     *(b)* FMPS     *(c)* FlowChef

*Figure 3.* Samples generated by different methods on MNIST for class-conditional sampling targeting class 3.

the significantly higher dimensionality of this dataset, additional diversification is required following the resampling phase. To address this, we propose to replace the ODE (11) with the SDE (12) over $[t_{n_2}, t_{n_2} + \delta]$ in Algorithm 1; the convergence properties of the algorithm are preserved under this modification, as justified by Proposition 3.2. We employ the CLIP model (Radford et al., 2021; Cherti et al., 2023) to specify the likelihood function and utilize an external attribute classifier for validation. As demonstrated in Table 3, our method achieves optimal $ACC_E$ performance while maintaining comparable CLIP scores. Generated samples are presented in Figure 5; further experimental details are deferred to Section F.4. Beyond text-to-image generation, we further demonstrate style-guided generation using the FLUX model, with qualitative results presented in Figure 7.

## 5. Related Work

Recent work has investigated training-free conditional sampling for flow matching models. One category of existing approaches attempts to minimize a cost function measuring the discrepancy between generated samples and desired properties by optimizing either the initial point or the solution trajectory of the ODE (Ben-Hamu et al., 2024; Wang et al., 2024; Patel et al., 2025). However, these optimization-based methods introduce bias between the sampling distribution and the target distribution $p_{\text{data}}(\cdot|y)$ (cf. Figures 2d and 2e), and tend to produce visually implausible images (cf. Figure 3c). A second category of approaches can be viewed as employing surrogate velocity fields to approximate the optimal conditional velocity field (Pokle et al., 2023; Feng

et al., 2025; Song et al., 2025). However, approximation error introduces a discrepancy between the sampling and target distributions (cf. Figure 2c), and the generated samples can exhibit sensitivity to random seeds in practice (cf. Figure 3b). In contrast, our method corrects approximation error through theoretically grounded sample weighting, thereby achieving asymptotic accuracy.

SMC has been explored in prior work for conditional sampling in score-based diffusion models (Trippe et al., 2022; Cardoso et al., 2023; Wu et al., 2023). Our method departs from them by building on flow matching models, whose generative dynamics are deterministic. This distinction motivates our key algorithmic contribution: an ODE-SDE-ODE sampling strategy that selectively introduces stochasticity at intermediate stages to facilitate resampling, while preserving deterministic dynamics at both ends.

This design offers two advantages over diffusion-based conditional samplers. First, whereas diffusion-based methods apply SDE simulation with resampling throughout the entire generation process, TFTF concentrates resampling within the intermediate stages where it is most beneficial. Specifically, early-phase samples are too noisy for reliable look-ahead discrimination, while in the late phase, coarse-scale structure has already been resolved, making resampling detrimental to sample diversity without commensurate gains in condition adherence. In both boundary phases where resampling is withheld, the stochasticity introduced by an SDE is superfluous; reverting to the ODE yields faster inference and lower numerical integration error. Second, existing diffusion-based methods are constrained to the pretrained model's noise schedule in their diffusion coefficient design, offering limited control over the degree of stochasticity. Our framework, by contrast, converts the deterministic flow of flow matching into a family of stochastic flows (12) governed by a tunable function $\alpha(t)$, whose scale can be increased to promote stronger exploratory behavior and improve intra-batch diversity.

As demonstrated in Figure 4, TFTF achieves superior classification accuracy and intra-batch diversity compared to TDS (Wu et al., 2023), corroborating the effectiveness of the proposed ODE-SDE-ODE strategy.

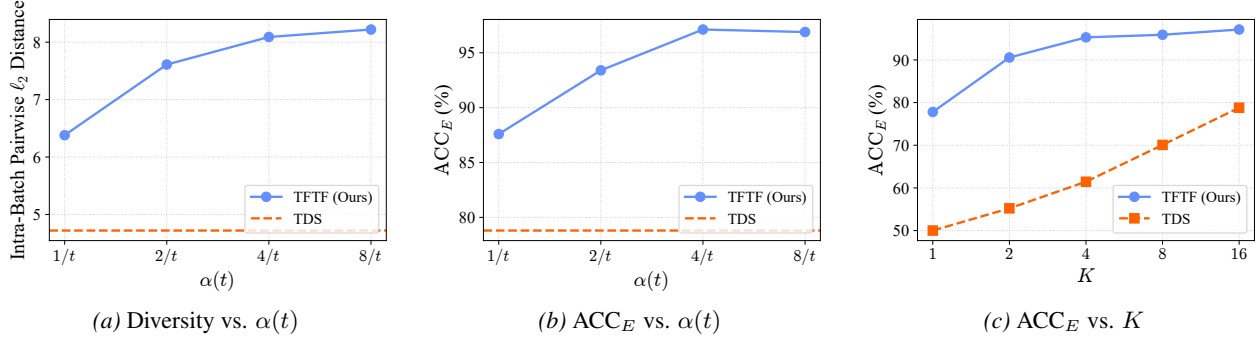

*(a)* Diversity vs. $\alpha(t)$      *(b)* $\mathrm{ACC}_E$ vs. $\alpha(t)$      *(c)* $\mathrm{ACC}_E$ vs. $K$

*Figure 4.* Ablation studies on the stochasticity parameter $\alpha(t)$ and particle count $K$.

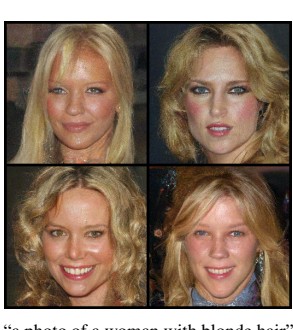

"a photo of a woman with blonde hair"

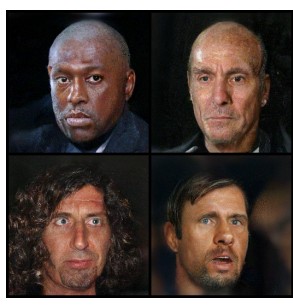

"a photo of a man with a serious expression"

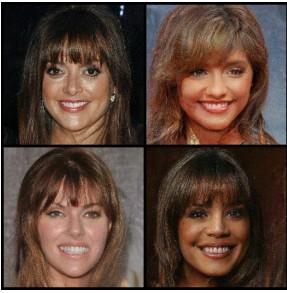

"a photo of a smiling woman with bangs"

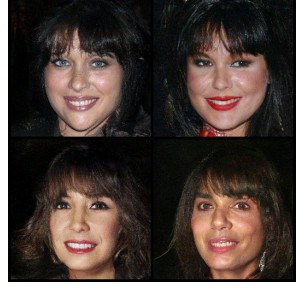

"a photo of a smiling woman with black hair and bangs"

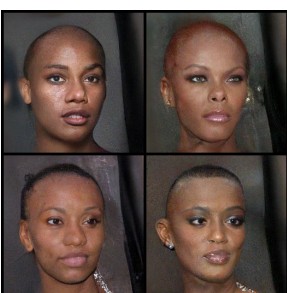

"a photo of a bald woman"

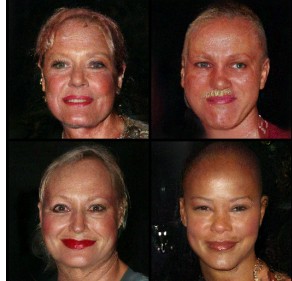

"a photo of a bald woman" (failure mode)

*Figure 5.* Samples generated by TFTF (Algorithm 1) with $K = 25$ for text-to-image generation. Each panel displays 4 samples drawn from the 25 weighted particles produced by TFTF. The bottom-right panel illustrates a failure mode observed when the text prompt is *"a photo of a bald woman"*: when the desired combination of attributes is rare under the unconditional distribution, gradient guidance via the surrogate velocity field, in conjunction with the enrichment effect of SMC resampling, may be insufficient to accurately capture the target condition under a finite particle budget.

## 6. Conclusion

We have presented a principled, training-free conditional sampling method for flow matching models based on importance sampling. By incorporating a resampling technique in SMC during intermediate stages, we enrich the particle set with promising partial samples and mitigate the weight degeneracy issue inherent to high-dimensional IS. The proposed framework is theoretically guaranteed to be asymptotically exact and has been empirically demonstrated to outperform existing approaches on conditional sampling tasks across several benchmark datasets.

Nevertheless, the present work is subject to certain limitations that warrant further investigation. First, the resampling interval $[t_{n_1}, t_{n_2}]$ in Algorithm 1 is a hyperparameter that currently requires manual specification via inspection of the pretrained model's generation trajectories. Converting this hyperparameter into an adaptive criterion is a promising direction for improvement; one potential approach is to monitor the variance of the look-ahead likelihood $\tilde{p}(y|\hat{x}_1(x(t)))$ as a proxy for whether the target feature is beginning to emerge, thereby enabling resampling to be triggered automatically. Second, while resampling at intermediate stages alleviates weight degeneracy, it simultaneously introduces inter-particle correlations, leading to a degree of visual similarity among images within the same batch (cf. Figures 5 and 9). This phenomenon, known in the SMC literature as path degeneracy, remains an open research problem. Finally, the present work focuses on image-based conditional generation in Euclidean space; extending the proposed methodology to Riemannian manifolds (for instance, to enable conditional generation of protein structures) constitutes an important direction for future work.

## Acknowledgements

The authors thank the anonymous referees for their insightful and constructive comments, which have led to an improved version of the paper. This work was supported by the Talent Introduction Research Start-up Fund of Tsinghua

University under grant number 53330000426, and by the Tsinghua University Disruptive Innovation Talent Development Program.

## Impact Statement

This paper presents work whose goal is to advance the field of Machine Learning. There are many potential societal consequences of our work, none which we feel must be specifically highlighted here.

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

*Table 4.* Summary of notation used throughout this paper.

| Notation | Description |
|---|---|
| $p_{\text{data}}(\cdot)$ | Unconditional data distribution |
| $p_{\text{data}}(\cdot \mid y)$ | Target conditional data distribution |
| $\tilde{p}(y \mid \cdot)$ | Time-independent likelihood function |
| $v_{\theta^*}(x, t)$ | Pretrained unconditional velocity field |
| $v_c^*(x(t), t)$ | Optimal conditional velocity field (8) |
| $v_c(x(t), t)$ | Practical conditional velocity field (10) |
| $f(\cdot, t)$ / $f_t(\cdot)$ | Marginal density under ODE (11) or equivalently SDE (12) |
| $h_{t_n}(\cdot \mid x_{t_{n-1}})$ | One-step conditional distribution of $X_2(t_n)$ given $X_2(t_{n-1}) = x_{t_{n-1}}$ under SDE (12) |
| $q_{t_n}(\cdot \mid x_{t_{n-1}})$ | One-step conditional distribution under SDE (12) with $v_{\theta^*}$ replaced by $v_c$; SMC proposal kernel |
| $\pi_{t_n}(\cdot)$ | Unnormalized intermediate target distribution in SMC |
| $\hat{x}_1(x(t))$ | One-step Euler projection of $x(t)$ to $t = 1$: $\hat{x}_1(x(t)) = x(t) + (1 - t)\, v_{\theta^*}(x(t), t)$ |
| $\alpha(t)$ | Nonnegative function controlling stochasticity level in SDE (12) |
| $\beta(t)$ | User-defined guidance scale in $v_c$ (10) |
| $[t_{n_1}, t_{n_2}]$ | Resampling interval in Algorithm 1, where $0 < t_{n_1} < t_{n_2} < 1$ |
| $K$ | Number of particles in Algorithm 1 |
| $\{(x_1^{(k)}, \bar{w}_1^{(k)})\}_{k=1}^K$ | Weighted samples returned by Algorithm 1 |

# A. Optimal Conditional Velocity Field

## A.1. Derivation of the Optimal Conditional Velocity Field

Below we demonstrate that the ordinary differential equation $dX(t) = v_c^*(X(t), t)dt$ transforms the initial noise distribution $\mathcal{N}(0, I)$ at $t = 0$ into the target distribution $p_{\text{data}}(\cdot|y)$ at $t = 1$, where the optimal conditional velocity field $v_c^*(x(t), t)$ takes the form $\mathbb{E}[\tilde{X}(1) - \tilde{X}(0)|\tilde{X}(t) = x(t), Y = y]$. The following proof adapts the approach of Liu et al. (2022a) to the conditional sampling setting.

Consider the stochastic process $\{\tilde{X}(t) = t\tilde{X}(1) + (1 - t)\tilde{X}(0) : t \in [0, 1]\}$ defined in Section 3. Conditioned on $Y = y$, we have $\tilde{X}(0)|Y = y \sim \mathcal{N}(0, I)$ and $\tilde{X}(1)|Y = y \sim p_{\text{data}}(\cdot|y)$. For any compactly supported continuously differentiable test function $h(\cdot) : \mathbb{R}^d \to \mathbb{R}$, it holds that

$$\frac{d\mathbb{E}[h(\tilde{X}(t))|Y = y]}{dt} = \mathbb{E}[\nabla h(\tilde{X}(t))^T \cdot (\tilde{X}(1) - \tilde{X}(0))|Y = y] = \mathbb{E}[\nabla h(\tilde{X}(t))^T \cdot v_c^*(\tilde{X}(t), t)|Y = y], \quad (20)$$

where the second equality follows from the law of iterated expectations.

Let $p(\cdot, t|y)$ denote the probability density function of $\tilde{X}(t)|Y = y$. From Equation (20), we obtain

$$\int h(x)\{\frac{\partial p(x, t|y)}{\partial t} + \nabla_x \cdot [v_c^*(x, t)p(x, t|y)]\}dx$$

$$= \int h(x)\frac{\partial p(x, t|y)}{\partial t}dx - \int \nabla h(x)^T \cdot v_c^*(x, t)p(x, t|y)dx$$

$$= \frac{d\mathbb{E}[h(\tilde{X}(t))|Y = y]}{dt} - \mathbb{E}[\nabla h(\tilde{X}(t))^T \cdot v_c^*(\tilde{X}(t), t)|Y = y] = 0, \quad (21)$$

where the first equality employs integration by parts.

By the arbitrariness of the test function $h(\cdot)$, we conclude that

$$\frac{\partial p(x, t|y)}{\partial t} = -\nabla_x \cdot [v_c^*(x, t)p(x, t|y)] \quad \text{almost everywhere.} \quad (22)$$

Note that Equation (22) coincides with the continuity equation (Øksendal, 2003) governing the probability evolution induced by the ODE

$$dX(t) = v_c^*(X(t), t)dt. \quad (23)$$

Consequently, we have established that the ODE $dX(t) = v_c^*(X(t), t)dt$ transforms $\mathcal{N}(0, I)$ at $t = 0$ into $p_{\text{data}}(\cdot|y)$ at $t = 1$ with the marginal density of $X(t)$ coinciding with $p(\cdot, t|y)$ for all $t \in [0, 1]$.

## A.2. Proof of Proposition 3.1

*Proof.* For notational convenience, we denote by $p_t(\cdot)$ the density function of $\tilde{X}(t)$ and by $p_t(\cdot|y)$ the density function of $\tilde{X}(t)|Y = y$.

By definition,

$$v_c^*(\tilde{x}(t), t) = \mathbb{E}[\tilde{X}(1) - \tilde{X}(0)|\tilde{X}(t) = \tilde{x}(t), Y = y]. \tag{24}$$

For $t \in (0, 1)$, exploiting the linear interpolation relationship $\tilde{X}(t) = t\tilde{X}(1) + (1 - t)\tilde{X}(0)$, we can express the equation as

$$
\begin{aligned}
v_c^*(\tilde{x}(t), t) &= \mathbb{E}[\tilde{X}(1) - \tilde{X}(0)|\tilde{X}(t) = \tilde{x}(t), Y = y] \\
&= \frac{\tilde{x}(t)}{t} + \frac{1 - t}{t} \cdot \mathbb{E}\left[-\frac{\tilde{X}(t) - t\tilde{X}(1)}{(1 - t)^2}\bigg|\tilde{X}(t) = \tilde{x}(t), Y = y\right] \\
&= \frac{\tilde{x}(t)}{t} + \frac{1 - t}{t} \cdot \int -\frac{\tilde{x}(t) - t\tilde{x}(1)}{(1 - t)^2} p(\tilde{x}(1)|\tilde{x}(t), y) \, d\tilde{x}(1).
\end{aligned} \tag{25}
$$

Invoking the conditional independence of $\tilde{X}(t)$ and $Y$ given $\tilde{X}(1)$, we can reformulate Equation (25) as

$$
\begin{aligned}
v_c^*(\tilde{x}(t), t) &= \frac{\tilde{x}(t)}{t} + \frac{1 - t}{t} \cdot \int -\frac{\tilde{x}(t) - t\tilde{x}(1)}{(1 - t)^2} p(\tilde{x}(1)|\tilde{x}(t), y) \, d\tilde{x}(1) \\
&= \frac{\tilde{x}(t)}{t} + \frac{1 - t}{t} \cdot \frac{\int -\frac{\tilde{x}(t) - t\tilde{x}(1)}{(1 - t)^2} p(\tilde{x}(t)|\tilde{x}(1)) p_1(\tilde{x}(1)|y) \, d\tilde{x}(1)}{p_t(\tilde{x}(t)|y)}.
\end{aligned} \tag{26}
$$

Noting that $\tilde{X}(t)|\tilde{X}(1) = \tilde{x}(1) \sim \mathcal{N}(t\tilde{x}(1), (1 - t)^2 I)$, it follows that $-\frac{\tilde{x}(t) - t\tilde{x}(1)}{(1 - t)^2} p(\tilde{x}(t)|\tilde{x}(1)) = \nabla_{\tilde{x}(t)} p(\tilde{x}(t)|\tilde{x}(1))$. Building upon this observation, we can continue the reformulation of Equation (26) to obtain

$$
\begin{aligned}
v_c^*(x(t), t) &= \frac{\tilde{x}(t)}{t} + \frac{1 - t}{t} \cdot \frac{\int -\frac{\tilde{x}(t) - t\tilde{x}(1)}{(1 - t)^2} p(\tilde{x}(t)|\tilde{x}(1)) p_1(\tilde{x}(1)|y) \, d\tilde{x}(1)}{p_t(\tilde{x}(t)|y)} \\
&= \frac{\tilde{x}(t)}{t} + \frac{1 - t}{t} \cdot \frac{\int \nabla_{\tilde{x}(t)} p(\tilde{x}(t)|\tilde{x}(1)) p_1(\tilde{x}(1)|y) \, d\tilde{x}(1)}{p_t(\tilde{x}(t)|y)} \\
&= \frac{\tilde{x}(t)}{t} + \frac{1 - t}{t} \cdot \frac{\nabla_{\tilde{x}(t)} \int p(\tilde{x}(t)|\tilde{x}(1)) p_1(\tilde{x}(1)|y) \, d\tilde{x}(1)}{p_t(\tilde{x}(t)|y)} \\
&= \frac{\tilde{x}(t)}{t} + \frac{1 - t}{t} \cdot \nabla_{\tilde{x}(t)} \log p_t(\tilde{x}(t)|y).
\end{aligned} \tag{27}
$$

Applying Bayes' theorem, we can rewrite Equation (27) as

$$
\begin{aligned}
v_c^*(\tilde{x}(t), t) &= \frac{\tilde{x}(t)}{t} + \frac{1 - t}{t} \cdot \nabla_{\tilde{x}(t)} \log p_t(\tilde{x}(t)|y) \\
&= \frac{\tilde{x}(t)}{t} + \frac{1 - t}{t} \cdot \nabla_{\tilde{x}(t)} \log p_t(\tilde{x}(t)) + \frac{1 - t}{t} \cdot \nabla_{\tilde{x}(t)} \log p_{Y|\tilde{X}(t)}(y|\tilde{x}(t)) \\
&= \frac{\tilde{x}(t)}{t} + \frac{1 - t}{t} \cdot \frac{\nabla_{\tilde{x}(t)} \int p(\tilde{x}(t)|\tilde{x}(1)) p_1(\tilde{x}(1)) \, d\tilde{x}(1)}{p_t(\tilde{x}(t))} + \frac{1 - t}{t} \cdot \nabla_{\tilde{x}(t)} \log p_{Y|\tilde{X}(t)}(y|\tilde{x}(t)) \\
&= \frac{\tilde{x}(t)}{t} + \frac{1 - t}{t} \cdot \frac{\int \nabla_{\tilde{x}(t)} p(\tilde{x}(t)|\tilde{x}(1)) p_1(\tilde{x}(1)) \, d\tilde{x}(1)}{p_t(\tilde{x}(t))} + \frac{1 - t}{t} \cdot \nabla_{\tilde{x}(t)} \log p_{Y|\tilde{X}(t)}(y|\tilde{x}(t)).
\end{aligned} \tag{28}
$$

Utilizing the identity $\nabla_{\tilde{x}(t)} p(\tilde{x}(t)|\tilde{x}(1)) = -\frac{\tilde{x}(t)-t\tilde{x}(1)}{(1-t)^2} p(\tilde{x}(t)|\tilde{x}(1))$, we can reformulate Equation (28) as

$$
\begin{aligned}
v_c^*(\tilde{x}(t), t) &= \frac{\tilde{x}(t)}{t} + \frac{1-t}{t} \cdot \frac{\int \nabla_{\tilde{x}(t)} p(\tilde{x}(t)|\tilde{x}(1)) p_1(\tilde{x}(1)) \, d\tilde{x}(1)}{p_t(\tilde{x}(t))} + \frac{1-t}{t} \cdot \nabla_{\tilde{x}(t)} \log p_{Y|\tilde{X}(t)}(y|\tilde{x}(t)) \\
&= \frac{\tilde{x}(t)}{t} + \frac{1-t}{t} \cdot \frac{\int -\frac{\tilde{x}(t)-t\tilde{x}(1)}{(1-t)^2} p(\tilde{x}(t)|\tilde{x}(1)) p_1(\tilde{x}(1)) \, d\tilde{x}(1)}{p_t(\tilde{x}(t))} + \frac{1-t}{t} \cdot \nabla_{\tilde{x}(t)} \log p_{Y|\tilde{X}(t)}(y|\tilde{x}(t)) \\
&= \frac{\tilde{x}(t)}{t} + \frac{1-t}{t} \cdot \int -\frac{\tilde{x}(t)-t\tilde{x}(1)}{(1-t)^2} p(\tilde{x}(1)|\tilde{x}(t)) \, d\tilde{x}(1) + \frac{1-t}{t} \cdot \nabla_{\tilde{x}(t)} \log p_{Y|\tilde{X}(t)}(y|\tilde{x}(t)) \\
&= \frac{\tilde{x}(t)}{t} + \frac{1-t}{t} \cdot \mathbb{E}\left[-\frac{\tilde{X}(t)-t\tilde{X}(1)}{(1-t)^2} \bigg| \tilde{X}(t) = \tilde{x}(t)\right] + \frac{1-t}{t} \cdot \nabla_{\tilde{x}(t)} \log p_{Y|\tilde{X}(t)}(y|\tilde{x}(t)) \\
&= \mathbb{E}[\tilde{X}(1) - \tilde{X}(0)|\tilde{X}(t) = \tilde{x}(t)] + \frac{1-t}{t} \cdot \nabla_{\tilde{x}(t)} \log p_{Y|\tilde{X}(t)}(y|\tilde{x}(t)) \\
&= v_{\theta^*}(\tilde{x}(t), t) + \frac{1-t}{t} \cdot \nabla_{\tilde{x}(t)} \log p_{Y|\tilde{X}(t)}(y|\tilde{x}(t)),
\end{aligned}
\tag{29}
$$

which is exactly the decomposition of the optimal conditional velocity field $v_c^*$ presented in Equation (8).

For $t = 1$, by the conditional independence of $\tilde{X}(0)$ and $Y$ given $\tilde{X}(1)$, we have

$$
\begin{aligned}
v_c^*(\tilde{x}(1), 1) &= \mathbb{E}[\tilde{X}(1) - \tilde{X}(0)|\tilde{X}(1) = \tilde{x}(1), Y = y] \\
&= \mathbb{E}[\tilde{X}(1) - \tilde{X}(0)|\tilde{X}(1) = \tilde{x}(1)] \\
&= v_{\theta^*}(\tilde{x}(1), 1),
\end{aligned}
\tag{30}
$$

which also satisfies Equation (8).

$\square$

## B. From Deterministic to Stochastic Flow

### B.1. Proof of Proposition 3.2

*Proof.* Consider the ordinary differential equation presented in Equation (11), which takes the form

$$
dX_1(t) = v_{\theta^*}(X_1(t), t)dt,
$$

with initial condition $X_1(0) \sim \mathcal{N}(0, I)$. In what follows, we employ the notation $f(\cdot, t)$ to denote the marginal probability density of $X_1(t)$, thereby representing its dependence on both state and time. According to the continuity equation (Øksendal, 2003), the marginal density evolves as follows:

$$
\frac{\partial f(x, t)}{\partial t} = -\nabla_x \cdot [v_{\theta^*}(x, t) f(x, t)].
\tag{31}
$$

To introduce stochasticity into the original ODE-based framework, we reformulate Equation (31) by adding zero in a strategically useful form:

$$
\begin{aligned}
\frac{\partial f(x, t)}{\partial t} &= -\nabla_x \cdot [v_{\theta^*}(x, t) f(x, t)] \\
&= -\nabla_x \cdot [v_{\theta^*}(x, t) f(x, t)] - (1-t)\alpha(t)\Delta_x f(x, t) + (1-t)\alpha(t)\Delta_x f(x, t) \\
&= -\nabla_x \cdot [v_{\theta^*}(x, t) f(x, t)] - (1-t)\alpha(t)\nabla_x \cdot [\nabla_x f(x, t)] + (1-t)\alpha(t)\Delta_x f(x, t) \\
&= -\nabla_x \cdot [v_{\theta^*}(x, t) f(x, t)] - (1-t)\alpha(t)\nabla_x \cdot [(\nabla_x \log f(x, t)) f(x, t)] + (1-t)\alpha(t)\Delta_x f(x, t) \\
&= -\nabla_x \cdot \{[v_{\theta^*}(x, t) + (1-t)\alpha(t)\nabla_x \log f(x, t)] f(x, t)\} + (1-t)\alpha(t)\Delta_x f(x, t),
\end{aligned}
\tag{32}
$$

where $\Delta_x$ denotes the Laplacian operator, defined as $\Delta_x f = \frac{\partial^2 f}{\partial x_1^2} + \frac{\partial^2 f}{\partial x_2^2} + \cdots + \frac{\partial^2 f}{\partial x_d^2}$.

For $t \in [0, 1)$, recall from Equation (2) that

$$f(\tilde{x}(t), t) = \int p(\tilde{x}(t)|\tilde{x}(1))p_{\text{data}}(\tilde{x}(1))d\tilde{x}(1),$$

where $p(\tilde{x}(t)|\tilde{x}(1)) = \mathcal{N}(\tilde{x}(t); t\tilde{x}(1), (1-t)^2 I)$. This yields $\nabla_{\tilde{x}(t)} p(\tilde{x}(t)|\tilde{x}(1)) = -\frac{\tilde{x}(t)-t\tilde{x}(1)}{(1-t)^2} p(\tilde{x}(t)|\tilde{x}(1))$. Consequently, we obtain

$$
\begin{aligned}
(1-t)\alpha(t)\nabla_{\tilde{x}(t)} \log f(\tilde{x}(t), t) &= \frac{(1-t)\alpha(t)\nabla_{\tilde{x}(t)} f(\tilde{x}(t), t)}{f(\tilde{x}(t), t)} \\
&= \frac{(1-t)\alpha(t)\nabla_{\tilde{x}(t)} \int p(\tilde{x}(t)|\tilde{x}(1))p_{\text{data}}(\tilde{x}(1))d\tilde{x}(1)}{f(\tilde{x}(t), t)} \\
&= \frac{(1-t)\alpha(t) \int \nabla_{\tilde{x}(t)} p(\tilde{x}(t)|\tilde{x}(1))p_{\text{data}}(\tilde{x}(1))d\tilde{x}(1)}{f(\tilde{x}(t), t)} \\
&= \frac{(1-t)\alpha(t) \int -\frac{\tilde{x}(t)-t\tilde{x}(1)}{(1-t)^2} p(\tilde{x}(t)|\tilde{x}(1))p_{\text{data}}(\tilde{x}(1))d\tilde{x}(1)}{f(\tilde{x}(t), t)} \\
&= \alpha(t) \int -\frac{\tilde{x}(t)-t\tilde{x}(1)}{1-t} p(\tilde{x}(1)|\tilde{x}(t))d\tilde{x}(1) \\
&= \alpha(t)\mathbb{E}\left[ -\frac{\tilde{X}(t)-t\tilde{X}(1)}{1-t} \bigg| \tilde{X}(t) = \tilde{x}(t) \right] \\
&= \alpha(t)\{-\tilde{x}(t) + t\mathbb{E}[\tilde{X}(1) - \tilde{X}(0)|\tilde{X}(t) = \tilde{x}(t)]\} \\
&= \alpha(t)[-\tilde{x}(t) + tv_{\theta^*}(\tilde{x}(t), t)].
\end{aligned}
\tag{33}
$$

For $t = 1$, we have

$$
\begin{aligned}
(1-t)\alpha(t)&\nabla_{\tilde{x}(t)} \log f(\tilde{x}(t), t)\big|_{t=1} = 0 \\
&= \alpha(1)\{-\tilde{x}(1) + \mathbb{E}\left[\tilde{X}(1) - \tilde{X}(0) \mid \tilde{X}(1) = \tilde{x}(1)\right]\} \\
&= \alpha(t)\left[-\tilde{x}(t) + tv_{\theta^*}(\tilde{x}(t), t)\right]\big|_{t=1}.
\end{aligned}
$$

By the above derivation, we can reformulate Equation (32) as

$$
\begin{aligned}
\frac{\partial f(x, t)}{\partial t} &= -\nabla_x \cdot \{[v_{\theta^*}(x, t) + (1-t)\alpha(t)\nabla_x \log f(x, t)]f(x, t)\} + (1-t)\alpha(t)\Delta_x f(x, t) \\
&= -\nabla_x \cdot \{[v_{\theta^*}(x, t) + \alpha(t)(-x + tv_{\theta^*}(x, t))]f(x, t)\} + (1-t)\alpha(t)\Delta_x f(x, t).
\end{aligned}
\tag{34}
$$

This partial differential equation corresponds to the Fokker-Planck-Kolmogorov equation (Särkkä & Solin, 2019) associated with the stochastic differential equation presented in Equation (12), which takes the form

$$dX_2(t) = \{\alpha(t)\left[-X_2(t) + tv_{\theta^*}(X_2(t), t)\right] + v_{\theta^*}(X_2(t), t)\} dt + \sqrt{2(1-t)\alpha(t)}\, dB(t).$$

Therefore, we have established that, given an initial distribution $\mathcal{N}(0, I)$, the stochastic processes corresponding to Equation (11) and Equation (12) share the same marginal distributions.

$\square$

## B.2. Connecting to Anderson's Reverse-Time SDE

Consider a stochastic process $\{X(t) : t \in [0, 1]\}$ modeled as the solution to the following SDE:

$$dX(t) = f(X(t), t)\, dt + g(t)\, d\bar{B}(t), \tag{35}$$

where $f : \mathbb{R}^d \times [0, 1] \to \mathbb{R}^d$ represents the drift coefficient, $g : \mathbb{R} \to \mathbb{R}$ represents the diffusion coefficient, $\bar{B}(t)$ denotes a standard Wiener process with time flowing backwards from 1 to 0, and the boundary condition is given by $X(1) \sim p_{\text{data}}$.

Anderson's seminal work (1982) establishes that this stochastic process can equivalently be characterized by the following SDE (known as Anderson's reverse-time SDE):

$$dX(t) = \left[ f(X(t), t) + g(t)^2 \nabla_{x(t)} \log p(X(t), t) \right] dt + g(t)\, dB(t), \tag{36}$$

where $B(t)$ denotes a standard Wiener process, and $p(\cdot, t)$ represents the marginal density of $X(t)$.

To establish the connection between the stochastic flow given by Equation (12) in Proposition 3.2 and Anderson's reverse-time SDE discussed above, we specify $f(x, t) = \frac{x}{t}$ and $g(t) = \sqrt{\frac{2(1-t)}{t}}$ in Equation (35). Since the drift coefficient $f$ is affine in $x$, the conditional distribution of $X(t)$ given $X(1)$ under Equation (35) can be derived analytically (see Chapter 5 of Särkkä & Solin, 2019):

$$p(x(t)|x(1)) = \mathcal{N}\left( x(t);\, tx(1),\, (1-t)^2 I \right). \tag{37}$$

Notably, this distribution coincides with the conditional distribution of $\tilde{X}(t)|\tilde{X}(1)$ specified in Equation (1). Consequently, the marginal probability density function $p(\cdot, t)$ of $X(t)$ is identical to the marginal density function $f(\cdot, t)$ of $\tilde{X}(t)$ given in Equation (2). For $t \in [0, 1)$, following a derivation analogous to that presented in Equation (33), the term $\nabla_{x(t)} \log p(x(t), t)$ in Equation (36) can therefore be computed as:

$$\nabla_{x(t)} \log p(x(t), t) = \nabla_{x(t)} \log f(x(t), t) = \frac{-x(t) + t v_\theta^*(x(t), t)}{1 - t}. \tag{38}$$

Substituting this expression, along with $f(x, t) = \frac{x}{t}$ and $g(t) = \sqrt{\frac{2(1-t)}{t}}$, into Equation (36) yields:

$$dX(t) = \left[ -\frac{X(t)}{t} + 2 v_\theta^*(X(t), t) \right] dt + \sqrt{\frac{2(1-t)}{t}}\, dB(t), \tag{39}$$

which is equivalent to setting $\alpha(t) = \frac{1}{t}$ in Equation (12). This demonstrates that Anderson's reverse-time SDE constitutes a special case of the stochastic flow given by Equation (12) presented in Proposition 3.2.

*Remark* B.1. At $t = 1$, we have $g(1) = 0$, which gives

$$\left[ f(X(t), t) + g(t)^2 \nabla_{X(t)} \log p(X(t), t) \right]\big|_{t=1} = X(1) = \left[ -\frac{X(t)}{t} + 2 v_\theta^*(X(t), t) \right]\bigg|_{t=1}.$$

*Remark* B.2. The functions $f(x, t) = \frac{x}{t}$ and $g(t) = \sqrt{\frac{2(1-t)}{t}}$ are singular at $t = 0$. To handle this rigorously, the preceding proof should be conducted on the interval $[\varepsilon, 1]$ for some $\varepsilon \in (0, 1)$. Accordingly, $\mathcal{N}(0, I)$ should be interpreted as the limiting distribution of the solution to the stochastic differential equation as $t \to 0^+$.

## C. Extended Target Distribution and Sample Reweighting at the Final Time Step

### C.1. Marginalizing the Final Extended Target Distribution

Below we demonstrate that the final extended target distribution contains the target distribution $p_{\text{data}}(x_1|y)$ as its marginal. As defined in Equation (15), the final extended target distribution is given by

$$f_{t_{n_1}}(x_{t_{n_1}}) \cdot \left[ \prod_{i=n_1+1}^{n_2} h_{t_i}(x_{t_i}|x_{t_{i-1}}) \right] \cdot e^{-\int_{t_{n_2}}^{1} \nabla \cdot v_{\theta^*}(x_t, t)\, dt} \cdot \tilde{p}(y|x_1), \tag{40}$$

where $h_{t_i}(x_{t_i}|x_{t_{i-1}})$ denotes the conditional distribution of $X_2(t_i)$ given $X_2(t_{i-1}) = x_{t_{i-1}}$ under SDE (12). By Proposition 3.2, the evolving marginal density governed by SDE (12) matches $f_t(\cdot)$ for all $t$, which yields

$$
\int f_{t_{n_1}}(x_{t_{n_1}}) \cdot \left[ \prod_{i=n_1+1}^{n_2} h_{t_i}(x_{t_i}|x_{t_{i-1}}) \right] \cdot e^{-\int_{t_{n_2}}^{1} \nabla \cdot v_{\theta^*}(x_t, t)\, dt} \cdot \tilde{p}(y|x_1)\, dx_{t_{n_1}:t_{n_2-1}}
$$

$$
= \int f_{t_{n_1}}(x_{t_{n_1}}) \cdot \left[ \prod_{i=n_1+1}^{n_2} h_{t_i}(x_{t_i}|x_{t_{i-1}}) \right] dx_{t_{n_1}:t_{n_2-1}} \cdot e^{-\int_{t_{n_2}}^{1} \nabla \cdot v_{\theta^*}(x_t, t)\, dt} \cdot \tilde{p}(y|x_1)
$$

$$
= f_{t_{n_2}}(x_{t_{n_2}}) \cdot e^{-\int_{t_{n_2}}^{1} \nabla \cdot v_{\theta^*}(x_t, t)\, dt} \cdot \tilde{p}(y|x_1) \tag{41}
$$

Since the ODE $dX(t) = v_{\theta^*}(X(t), t)dt$ transforms the marginal density from $f_{t_{n_2}}(\cdot)$ to $p_{\text{data}}(\cdot)$ over $[t_{n_2}, 1]$, applying Equation (7) yields

$$
f_{t_{n_2}}(x_{t_{n_2}}) \cdot e^{-\int_{t_{n_2}}^{1} \nabla \cdot v_{\theta^*}(x_t, t)\, dt} \cdot \tilde{p}(y|x_1) = p_{\text{data}}(x_1) \cdot \tilde{p}(y|x_1), \tag{42}
$$

which is proportional to the target distribution $p_{\text{data}}(x_1|y)$.

## C.2. Derivation of the Final Incremental Weight

Below we derive the final incremental weight given in Equation (17). At the terminal step of the intermediate stage in Algorithm 1, we obtain a collection of weighted samples $\{(x_{t_{n_1}:t_{n_2}}^{(k)}, \bar{w}_{t_{n_2}}^{(k)})\}_{k=1}^{K}$ that are properly weighted with respect to the intermediate target distribution at time $t = t_{n_2}$:

$$
\pi_{t_{n_2}}(x_{t_{n_1}:t_{n_2}}) = f_{t_{n_1}}(x_{t_{n_1}}) \cdot \left[ \prod_{i=n_1+1}^{n_2} h_{t_i}(x_{t_i} \mid x_{t_{i-1}}) \right] \cdot \tilde{p}(y \mid \hat{x}_1(x_{t_{n_2}})). \tag{43}
$$

Over the interval $[t_{n_2}, 1]$, we solve the ordinary differential equation $dx_t = v_{\theta^*}(x_t, t)\, dt$ from $t = t_{n_2}$ to $t = 1$ to obtain $x_1$. This ODE integration induces a change of variables transformation

$$
T : \mathbb{R}^{d \cdot (n_2 - n_1 + 1)} \to \mathbb{R}^{d \cdot (n_2 - n_1 + 1)}, \quad (x_{t_{n_1}}, \dots, x_{t_{n_2-1}}, x_{t_{n_2}}) \mapsto (x_{t_{n_1}}, \dots, x_{t_{n_2-1}}, x_1), \quad x_{t_i} \in \mathbb{R}^d. \tag{44}
$$

Applying this transformation to each sample yields

$$
(x_{t_{n_1}:t_{n_2-1}}, x_1)^{(k)} = T(x_{t_{n_1}:t_{n_2}}^{(k)}), \quad k = 1, \dots, K. \tag{45}
$$

By the change of variables formula, the transformed samples $\{(T(x_{t_{n_1}:t_{n_2}}^{(k)}), \bar{w}_{t_{n_2}}^{(k)})\}_{k=1}^{K}$ are properly weighted with respect to $\pi_{t_{n_2}}(x_{t_{n_1}:t_{n_2}}) \cdot |\det J_T|^{-1}$, where $J_T$ denotes the Jacobian matrix of $T$. Since the transformation $T$ leaves the first $n_2 - n_1$ dimensions unchanged, we have

$$
|\det J_T|^{-1} = \left| \det \frac{dx_1}{dx_{t_{n_2}}} \right|^{-1}. \tag{46}
$$

Applying Equation (7) yields

$$
\left| \det \frac{dx_1}{dx_{t_{n_2}}} \right|^{-1} = e^{-\int_{t_{n_2}}^{1} \nabla \cdot v_{\theta^*}(x_t, t)\, dt}. \tag{47}
$$

Consequently, the samples $\{(T(x_{t_{n_1}:t_{n_2}}^{(k)}), \bar{w}_{t_{n_2}}^{(k)})\}_{k=1}^{K}$ are properly weighted with respect to

$$
\pi_{t_{n_2}}(x_{t_{n_1}:t_{n_2}}) \cdot e^{-\int_{t_{n_2}}^{1} \nabla \cdot v_{\theta^*}(x_t, t)\, dt}. \tag{48}
$$

To correct for the discrepancy between this distribution and the desired final extended target distribution given in Equation (15), the final incremental weight should be computed as

$$
\frac{f_{t_{n_1}}(x_{t_{n_1}}) \cdot \left[\prod_{i=n_1+1}^{n_2} h_{t_i}(x_{t_i}|x_{t_{i-1}})\right] \cdot e^{-\int_{t_{n_2}}^{1} \nabla \cdot v_{\theta^*}(x_t,t)\,dt} \cdot \tilde{p}(y|x_1)}{\pi_{t_{n_2}}(x_{t_{n_1}:t_{n_2}}) \cdot e^{-\int_{t_{n_2}}^{1} \nabla \cdot v_{\theta^*}(x_t,t)\,dt}}
$$

$$
= \frac{f_{t_{n_1}}(x_{t_{n_1}}) \cdot \left[\prod_{i=n_1+1}^{n_2} h_{t_i}(x_{t_i}|x_{t_{i-1}})\right] \cdot \tilde{p}(y|x_1)}{f_{t_{n_1}}(x_{t_{n_1}}) \cdot \left[\prod_{i=n_1+1}^{n_2} h_{t_i}(x_{t_i}|x_{t_{i-1}})\right] \cdot \tilde{p}(y|\hat{x}_1(x_{t_{n_2}}))}
$$

$$
= \frac{\tilde{p}(y|x_1)}{\tilde{p}(y|\hat{x}_1(x_{t_{n_2}}))}, \tag{49}
$$

which is precisely the final incremental weight specified in Equation (17). We verify in Section D.1 that this sample reweighting formula yields weighted samples that converge to the target distribution $p_{\text{data}}(\cdot|y)$ asymptotically.

# D. Asymptotic Accuracy of Algorithm 1

## D.1. Proof of Proposition 3.3

*Proof.* We regard the process of solving the ODE $dx_t = v_{\theta^*}(x_t, t)\,dt$ on the interval $[t_{n_2}, 1]$ in Algorithm 1 as a deterministic transformation

$$
T^* : \mathbb{R}^d \to \mathbb{R}^d, \quad x_{t_{n_2}} \mapsto x_1.
$$

Substituting into the expression for the final incremental weight, we obtain

$$
\sum_{k=1}^{K} \bar{W}_1^{(k)} \phi(X_1^{(k)}) = \frac{\sum_{k=1}^{K} W_1^{(k)} \phi(X_1^{(k)})}{\sum_{k=1}^{K} W_1^{(k)}}
$$

$$
= \frac{\sum_{k=1}^{K} \bar{W}_{t_{n_2}}^{(k)} \frac{\tilde{p}(y|T^*(X_{t_{n_2}}^{(k)}))}{\tilde{p}(y|\hat{x}_1(X_{t_{n_2}}^{(k)}))} \phi(T^*(X_{t_{n_2}}^{(k)}))}{\sum_{k=1}^{K} \bar{W}_{t_{n_2}}^{(k)} \frac{\tilde{p}(y|T^*(X_{t_{n_2}}^{(k)}))}{\tilde{p}(y|\hat{x}_1(X_{t_{n_2}}^{(k)}))}}
$$

$$
= \frac{\sum_{k=1}^{K} \bar{W}_{t_{n_2}}^{(k)} \phi_1(X_{t_{n_2}}^{(k)})}{\sum_{k=1}^{K} \bar{W}_{t_{n_2}}^{(k)} \phi_2(X_{t_{n_2}}^{(k)})}, \tag{50}
$$

where we define

$$
\phi_1(x_{t_{n_2}}) := \frac{\tilde{p}(y|T^*(x_{t_{n_2}}))}{\tilde{p}(y|\hat{x}_1(x_{t_{n_2}}))} \phi(T^*(x_{t_{n_2}})), \quad \phi_2(x_{t_{n_2}}) := \frac{\tilde{p}(y|T^*(x_{t_{n_2}}))}{\tilde{p}(y|\hat{x}_1(x_{t_{n_2}}))}.
$$

Under the continuity assumptions on the likelihood function $\tilde{p}(y|\cdot)$ and the transformation $T^*$ (which hold under general statistical model assumptions and neural network architectures), both $\phi_1$ and $\phi_2$ belong to $\mathcal{C}_b(\mathbb{R}^d)$.

As specified in Equation (13), the intermediate target distribution at $t = t_{n_2}$ is given by

$$
\pi_{t_{n_2}}(x_{t_{n_1}:t_{n_2}}) = f_{t_{n_1}}(x_{t_{n_1}}) \cdot \left[\prod_{i=n_1+1}^{n_2} h_{t_i}(x_{t_i}|x_{t_{i-1}})\right] \cdot \tilde{p}(y|\hat{x}_1(x_{t_{n_2}})),
$$

where $h_{t_i}(x_{t_i}|x_{t_{i-1}})$ denotes the conditional distribution of $X_2(t_i)$ given $X_2(t_{i-1}) = x_{t_{i-1}}$ under SDE (12). By Proposition 3.2, the evolving marginal density governed by SDE (12) coincides with $f_t(\cdot)$ for all $t$. Consequently, integrating out $x_{t_{n_1}:t_{n_2-1}}$ yields the unnormalized marginal density with respect to $x_{t_{n_2}}$:

$$
\int \pi_{t_{n_2}}(x_{t_{n_1}:t_{n_2}})\,dx_{t_{n_1}:t_{n_2-1}} = \int f_{t_{n_1}}(x_{t_{n_1}}) \cdot \left[\prod_{i=n_1+1}^{n_2} h_{t_i}(x_{t_i}|x_{t_{i-1}})\right] \cdot \tilde{p}(y|\hat{x}_1(x_{t_{n_2}}))\,dx_{t_{n_1}:t_{n_2-1}}
$$

$$
= f_{t_{n_2}}(x_{t_{n_2}})\tilde{p}(y|\hat{x}_1(x_{t_{n_2}})). \tag{51}
$$

We denote the corresponding normalized marginal density by

$$\pi^*_{t_{n_2}}(x_{t_{n_2}}) := \frac{f_{t_{n_2}}(x_{t_{n_2}})\tilde{p}(y|\hat{x}_1(x_{t_{n_2}}))}{\int f_{t_{n_2}}(x_{t_{n_2}})\tilde{p}(y|\hat{x}_1(x_{t_{n_2}}))\,dx_{t_{n_2}}}. \tag{52}$$

Since $\phi_1, \phi_2 \in \mathcal{C}_b(\mathbb{R}^d)$, by Chopin et al. (2020, Proposition 11.4), as $K \to \infty$,

$$\sum_{k=1}^{K} \bar{W}^{(k)}_{t_{n_2}} \phi_1(X^{(k)}_{t_{n_2}}) \xrightarrow{\text{a.s.}} \int \phi_1(x_{t_{n_2}})\pi^*_{t_{n_2}}(x_{t_{n_2}})\,dx_{t_{n_2}}, \tag{53}$$

$$\sum_{k=1}^{K} \bar{W}^{(k)}_{t_{n_2}} \phi_2(X^{(k)}_{t_{n_2}}) \xrightarrow{\text{a.s.}} \int \phi_2(x_{t_{n_2}})\pi^*_{t_{n_2}}(x_{t_{n_2}})\,dx_{t_{n_2}}. \tag{54}$$

It follows that the ratio converges as well:

$$\frac{\sum_{k=1}^{K} \bar{W}^{(k)}_{t_{n_2}} \phi_1(X^{(k)}_{t_{n_2}})}{\sum_{k=1}^{K} \bar{W}^{(k)}_{t_{n_2}} \phi_2(X^{(k)}_{t_{n_2}})} \xrightarrow{\text{a.s.}} \frac{\int \phi_1(x_{t_{n_2}})\pi^*_{t_{n_2}}(x_{t_{n_2}})\,dx_{t_{n_2}}}{\int \phi_2(x_{t_{n_2}})\pi^*_{t_{n_2}}(x_{t_{n_2}})\,dx_{t_{n_2}}}. \tag{55}$$

It remains to show that the limiting ratio equals $\mathbb{E}_{p_{\text{data}}(\cdot|y)}[\phi]$. Substituting the definitions of $\phi_1$, $\phi_2$, and $\pi^*_{t_{n_2}}$, we have

$$\frac{\int \phi_1(x_{t_{n_2}})\pi^*_{t_{n_2}}(x_{t_{n_2}})\,dx_{t_{n_2}}}{\int \phi_2(x_{t_{n_2}})\pi^*_{t_{n_2}}(x_{t_{n_2}})\,dx_{t_{n_2}}}$$

$$= \frac{\int \frac{\tilde{p}(y|T^*(x_{t_{n_2}}))}{\tilde{p}(y|\hat{x}_1(x_{t_{n_2}}))}\phi(T^*(x_{t_{n_2}}))f_{t_{n_2}}(x_{t_{n_2}})\tilde{p}(y|\hat{x}_1(x_{t_{n_2}}))\,dx_{t_{n_2}}}{\int \frac{\tilde{p}(y|T^*(x_{t_{n_2}}))}{\tilde{p}(y|\hat{x}_1(x_{t_{n_2}}))}f_{t_{n_2}}(x_{t_{n_2}})\tilde{p}(y|\hat{x}_1(x_{t_{n_2}}))\,dx_{t_{n_2}}}$$

$$= \frac{\int \tilde{p}(y|T^*(x_{t_{n_2}}))\phi(T^*(x_{t_{n_2}}))f_{t_{n_2}}(x_{t_{n_2}})\,dx_{t_{n_2}}}{\int \tilde{p}(y|T^*(x_{t_{n_2}}))f_{t_{n_2}}(x_{t_{n_2}})\,dx_{t_{n_2}}}. \tag{56}$$

Finally, we perform the change of variables $x_1 = T^*(x_{t_{n_2}})$, which allows us to rewrite Equation (56) as

$$\frac{\int \tilde{p}(y|T^*(x_{t_{n_2}}))\phi(T^*(x_{t_{n_2}}))f_{t_{n_2}}(x_{t_{n_2}})\,dx_{t_{n_2}}}{\int \tilde{p}(y|T^*(x_{t_{n_2}}))f_{t_{n_2}}(x_{t_{n_2}})\,dx_{t_{n_2}}} = \frac{\int \tilde{p}(y|x_1)\phi(x_1)p_{\text{data}}(x_1)\,dx_1}{\int \tilde{p}(y|x_1)p_{\text{data}}(x_1)\,dx_1}$$

$$= \int \phi(x_1)p_{\text{data}}(x_1|y)\,dx_1$$

$$= \mathbb{E}_{p_{\text{data}}(\cdot|y)}[\phi], \tag{57}$$

where the second equality follows from Bayes' theorem. This completes the proof. $\square$

## D.2. Implementation Details and Verification of Boundedness Conditions

This section first describes the implementation of Algorithm 1, then verifies that this implementation satisfies the boundedness conditions required by Proposition 3.3 (namely, the intermediate weight functions as well as the final increment weight are bounded).

As stated in Section 3.2, for $t_{n_1} \le t \le t_{n_2}$, we set the intermediate target distribution $\pi_{t_n}(x_{t_{n_1}:t_n})$ to be

$$f_{t_{n_1}}(x_{t_{n_1}})\left[\prod_{i=n_1+1}^{n} h_{t_i}(x_{t_i}|x_{t_{i-1}})\right]\tilde{p}(y|\hat{x}_1(x_{t_n})), \tag{58}$$

where $h_{t_i}(x_{t_i}|x_{t_{i-1}})$ denotes the conditional distribution of $X_2(t_i)$ given $X_2(t_{i-1}) = x_{t_{i-1}}$ under SDE (12). The proposal distribution at $t = t_{n_1}$ is set to $f_{t_{n_1}}(x_{t_{n_1}})$. For the subsequent proposal kernels, we first define $v_c$ as in Equation (10):

$$v_c(x(t), t) = v_{\theta^*}(x(t), t) + \beta(t) \cdot \frac{1-t}{t} \cdot \nabla_{x(t)} \log \tilde{p}(y|\hat{x}_1(x(t))).$$

The proposal kernel $q_{t_n}(x_{t_n}|x_{t_{n-1}})$ for $n_1 < n \leq n_2$ is then defined as the conditional distribution of $X_2(t_n)$ given $X_2(t_{n-1}) = x_{t_{n-1}}$ under SDE (12), with $v_{\theta^*}$ replaced by $v_c$.

In our implementation, we compute $h_{t_n}(x_{t_n}|x_{t_{n-1}})$ using the Euler–Maruyama discretization:

$$h_{t_n}(x_{t_n}|x_{t_{n-1}}) \approx \mathcal{N}(x_{t_n}; \mu_{t_{n-1}}(x_{t_{n-1}}), \sigma^2_{t_{n-1}}I), \tag{59}$$

where

$$\mu_{t_{n-1}}(x_{t_{n-1}}) := x_{t_{n-1}} + \{\alpha(t_{n-1})\left[-x_{t_{n-1}} + t_{n-1}v_{\theta^*}(x_{t_{n-1}}, t_{n-1})\right] + v_{\theta^*}(x_{t_{n-1}}, t_{n-1})\}(t_n - t_{n-1}), \tag{60}$$

$$\sigma^2_{t_{n-1}} := 2(1 - t_{n-1})\alpha(t_{n-1})(t_n - t_{n-1}). \tag{61}$$

Similarly, we compute $q_{t_n}(x_{t_n}|x_{t_{n-1}})$ as

$$q_{t_n}(x_{t_n}|x_{t_{n-1}}) \approx \mathcal{N}(x_{t_n}; \tilde{\mu}_{t_{n-1}}(x_{t_{n-1}}), \tilde{\sigma}^2_{t_{n-1}}I), \tag{62}$$

where

$$\tilde{\mu}_{t_{n-1}}(x_{t_{n-1}}) := x_{t_{n-1}} + \{\alpha(t_{n-1})\left[-x_{t_{n-1}} + t_{n-1}v_c(x_{t_{n-1}}, t_{n-1})\right] + v_c(x_{t_{n-1}}, t_{n-1})\}(t_n - t_{n-1}), \tag{63}$$

$$\tilde{\sigma}^2_{t_{n-1}} := \sigma^2_{t_{n-1}} + \epsilon_1, \quad \text{with} \quad 0 < \epsilon_1 \ll 1. \tag{64}$$

To ensure that the boundedness conditions of Proposition 3.3 are satisfied, we apply three measures: first, a small constant $0 < \epsilon_1 \ll 1$ is added to the proposal kernel variance $\tilde{\sigma}^2_{t_{n-1}}$, as shown in Equation (64); second, a small constant $0 < \epsilon_2 \ll 1$ is added to $\tilde{p}(y|\hat{x}_1(x_{t_n}))$ for $t_{n_1} \leq t \leq t_{n_2}$ (note that no such constant is added in the computation of $\tilde{p}(y|x_1)$); third, the functions $\alpha(t)$ and $\beta(t)$ are chosen to be bounded on the interval $[t_{n_1}, t_{n_2}]$, and norm clipping is applied to the gradient term $\nabla_{x(t)} \log \tilde{p}(y|\hat{x}_1(x(t)))$ in $v_c$.

Importantly, these modifications do not affect the asymptotic accuracy of Algorithm 1 for sampling from the target distribution $p_{\text{data}}(x_1|y) \propto p_{\text{data}}(x_1)\tilde{p}(y|x_1)$. To verify this, one need only replace $\tilde{p}(y|\hat{x}_1(X^{(k)}_{t_{n_2}}))$ with $\tilde{p}(y|\hat{x}_1(X^{(k)}_{t_{n_2}})) + \epsilon_2$ in the proof presented in Section D.1.

Under this implementation, the intermediate weight functions are given by

$$w_{t_n} = \begin{cases} \tilde{p}(y|\hat{x}_1(x_{t_{n_1}})) + \epsilon_2, & \text{for } n = n_1, \\ \dfrac{(\tilde{p}(y|\hat{x}_1(x_{t_n})) + \epsilon_2)\, h(x_{t_n}|x_{t_{n-1}})}{(\tilde{p}(y|\hat{x}_1(x_{t_{n-1}})) + \epsilon_2)\, q(x_{t_n}|x_{t_{n-1}})}, & \text{for } n_1 < n \leq n_2. \end{cases} \tag{65}$$

The final increment weight is given by

$$\frac{\tilde{p}(y|x_1)}{\tilde{p}(y|\hat{x}_1(x_{t_{n_2}})) + \epsilon_2}. \tag{66}$$

We now verify that the intermediate weight functions (65) and the final increment weight (66) are bounded. According to Equation (59) and Equation (62), we have

$$\log \frac{h(x_{t_n}|x_{t_{n-1}})}{q(x_{t_n}|x_{t_{n-1}})} = -\frac{1}{2}\left[\sigma^{-2}_{t_{n-1}}\|x_{t_n} - \mu_{t_{n-1}}(x_{t_{n-1}})\|^2 - \tilde{\sigma}^{-2}_{t_{n-1}}\|x_{t_n} - \tilde{\mu}_{t_{n-1}}(x_{t_{n-1}})\|^2\right] + \log \frac{|2\pi\sigma^2_{t_{n-1}}I|^{-1/2}}{|2\pi\tilde{\sigma}^2_{t_{n-1}}I|^{-1/2}}. \tag{67}$$

Applying the identity

$$\begin{aligned} \|x_{t_n} - \tilde{\mu}_{t_{n-1}}(x_{t_{n-1}})\|^2 &= \|x_{t_n} - \mu_{t_{n-1}}(x_{t_{n-1}})\|^2 \\ &+ 2\langle x_{t_n} - \mu_{t_{n-1}}(x_{t_{n-1}}), \mu_{t_{n-1}}(x_{t_{n-1}}) - \tilde{\mu}_{t_{n-1}}(x_{t_{n-1}})\rangle \\ &+ \|\mu_{t_{n-1}}(x_{t_{n-1}}) - \tilde{\mu}_{t_{n-1}}(x_{t_{n-1}})\|^2 \end{aligned} \tag{68}$$

yields

$$\begin{aligned} \sigma^{-2}_{t_{n-1}}&\|x_{t_n} - \mu_{t_{n-1}}(x_{t_{n-1}})\|^2 - \tilde{\sigma}^{-2}_{t_{n-1}}\|x_{t_n} - \tilde{\mu}_{t_{n-1}}(x_{t_{n-1}})\|^2 \\ &= \left(\sigma^{-2}_{t_{n-1}} - \tilde{\sigma}^{-2}_{t_{n-1}}\right)\|x_{t_n} - \mu_{t_{n-1}}(x_{t_{n-1}})\|^2 \\ &- 2\tilde{\sigma}^{-2}_{t_{n-1}}\langle x_{t_n} - \mu_{t_{n-1}}(x_{t_{n-1}}), \mu_{t_{n-1}}(x_{t_{n-1}}) - \tilde{\mu}_{t_{n-1}}(x_{t_{n-1}})\rangle \\ &- \tilde{\sigma}^{-2}_{t_{n-1}}\|\mu_{t_{n-1}}(x_{t_{n-1}}) - \tilde{\mu}_{t_{n-1}}(x_{t_{n-1}})\|^2. \end{aligned} \tag{69}$$

Applying the Cauchy–Schwarz inequality, we obtain

$$
\begin{aligned}
&\sigma_{t_{n-1}}^{-2}\|x_{t_n} - \mu_{t_{n-1}}(x_{t_{n-1}})\|^2 - \tilde{\sigma}_{t_{n-1}}^{-2}\|x_{t_n} - \tilde{\mu}_{t_{n-1}}(x_{t_{n-1}})\|^2 \\
&\leq \left(\sigma_{t_{n-1}}^{-2} - \tilde{\sigma}_{t_{n-1}}^{-2}\right)\|x_{t_n} - \mu_{t_{n-1}}(x_{t_{n-1}})\|^2 \\
&\quad + 2\tilde{\sigma}_{t_{n-1}}^{-2}\|x_{t_n} - \mu_{t_{n-1}}(x_{t_{n-1}})\| \cdot \|\mu_{t_{n-1}}(x_{t_{n-1}}) - \tilde{\mu}_{t_{n-1}}(x_{t_{n-1}})\| \\
&\quad - \tilde{\sigma}_{t_{n-1}}^{-2}\|\mu_{t_{n-1}}(x_{t_{n-1}}) - \tilde{\mu}_{t_{n-1}}(x_{t_{n-1}})\|^2.
\end{aligned}
\tag{70}
$$

This expression is bounded from below for the following two reasons: first, the quadratic function in $\|x_{t_n} - \mu_{t_{n-1}}(x_{t_{n-1}})\|$:

$$
\left(\sigma_{t_{n-1}}^{-2} - \tilde{\sigma}_{t_{n-1}}^{-2}\right)\|x_{t_n} - \mu_{t_{n-1}}(x_{t_{n-1}})\|^2 + 2\tilde{\sigma}_{t_{n-1}}^{-2}\|x_{t_n} - \mu_{t_{n-1}}(x_{t_{n-1}})\| \cdot \|\mu_{t_{n-1}}(x_{t_{n-1}}) - \tilde{\mu}_{t_{n-1}}(x_{t_{n-1}})\| \tag{71}
$$

has a positive leading coefficient (since $\sigma_{t_{n-1}}^{-2} > \tilde{\sigma}_{t_{n-1}}^{-2}$ due to the addition of $\epsilon_1$) and therefore admits a lower bound; second, the norm of the mean difference:

$$
\|\mu_{t_{n-1}}(x_{t_{n-1}}) - \tilde{\mu}_{t_{n-1}}(x_{t_{n-1}})\| = \|[\alpha(t_{n-1})t_{n-1} + 1] \cdot (t_n - t_{n-1}) \cdot \beta(t_{n-1}) \cdot \frac{1 - t_{n-1}}{t_{n-1}} \cdot \nabla_{x(t)} \log \tilde{p}(y|\hat{x}_1(x(t)))\| \tag{72}
$$

is bounded due to the norm clipping applied to the gradient term $\nabla_{x_{t_{n-1}}} \log \tilde{p}(y|\hat{x}_1(x_{t_{n-1}}))$. It follows that Equation (67) is upper-bounded and thus $\frac{h(x_{t_n}|x_{t_{n-1}})}{q(x_{t_n}|x_{t_{n-1}})}$ is bounded.

Since the likelihood function $\tilde{p}(y|\cdot)$ is nonnegative and bounded in typical conditional sampling scenarios, we conclude that both the intermediate weight functions (65) and the final increment weight (66) are bounded. This verifies that the boundness conditions required by Proposition 3.3 are satisfied.

# E. Distributed Implementation of TFTF

## E.1. Nested Training-Free Targeted Flow

As presented in Algorithm 2, we adopt a nested structure following Vergé et al. (2015) for distributed sampling. Specifically, the inner level follows the implementation of Algorithm 1 within each node, while the outer level performs reweighting and resampling across nodes. To reduce inter-node communication, we employ adaptive resampling at the outer level: resampling is triggered only when the effective sample size corresponding to the node weights falls below a specified threshold; otherwise, weights are simply accumulated without resampling. (In practice, the $M$ nodes may represent distributed computational nodes, multi-GPU setups, or $M$ sequential executions on a single device.)

A key property of Algorithm 2 is that, for a fixed per-node sample size, the resulting weighted samples converge to the target distribution as the number of nodes increases, as stated in the following proposition.

**Proposition E.1** (Proof in Section E.2). *Consider Algorithm 2 under the same conditions as Algorithm 1. Assuming resampling is performed at every outer-level iteration, for any fixed $K \in \mathbb{N}_+$ and any $\phi \in \mathcal{C}_b(\mathbb{R}^d)$, we have*

$$
\sum_{m=1}^{M}\sum_{k=1}^{K} \bar{W}_1^{(m,k)}\phi(X_1^{(m,k)}) \xrightarrow{\text{a.s.}} \mathbb{E}_{p_{\text{data}}(\cdot|y)}[\phi] \quad \text{as } M \to \infty. \tag{73}
$$

*Remark* E.2. While Algorithm 2 employs ESS-based adaptive resampling at the outer level to reduce inter-node communication, the theoretical guarantee in Proposition E.1 is established under the assumption that outer-level resampling is performed at every iteration. Extending convergence results to ESS-based adaptive resampling schemes remains an open problem in SMC theory; see Del Moral et al. (2012) for progress in this direction.

To demonstrate the asymptotic accuracy of Algorithm 2, we revisit the toy example from Section 4.1, where the target conditional distribution $p_{\text{data}}(\cdot|y)$ is a two-dimensional Gaussian mixture: $\frac{3}{4}\mathcal{N}((-\frac{\sqrt{2}}{2}, -\frac{\sqrt{2}}{2}), \frac{1}{8}I) + \frac{1}{4}\mathcal{N}((\frac{\sqrt{2}}{2}, \frac{\sqrt{2}}{2}), \frac{1}{8}I)$. We fix $K = 4$ in Algorithm 2 and measure the Wasserstein-2 distance between our generated samples and i.i.d. ground-truth samples as a function of $M$. As illustrated in Figure 6, the Wasserstein-2 distance decreases progressively as the number of nodes $M$ increases. This demonstrates that the cross-node reweighting and resampling procedure employed by Algorithm 2 effectively mitigates the sampling bias arising from the finite sample sizes at individual nodes.

---

**Algorithm 2** Nested Training-Free Targeted Flow (Nested TFTF)

---

**Require:** Pretrained velocity field $v_{\theta^*}$, likelihood function $\tilde{p}(y|\cdot)$, intermediate proposal distributions $\{q_{t_n}\}_{n=n_1}^{n_2}$, intermediate target distributions $\{\pi_{t_n}\}_{n=n_1}^{n_2}$, number of nodes $M$, outer resampling threshold $M_\tau$, number of samples per node $K$

1: **Step 1: Initialization**
2: **for** $m = 1, \ldots, M$ **do**
3:    **for** $k = 1, \ldots, K$ **do**
4:       Sample $x_0^{(m,k)} \sim \mathcal{N}(0, I)$
5:       Solve $\mathrm{d}x_t = v_{\theta^*}(x_t, t)\mathrm{d}t$ from $t = 0$ to $t = t_{n_1}$ to obtain $x_{t_{n_1}}^{(m,k)}$
6:       Compute $w_{t_{n_1}}^{(m,k)} = \dfrac{\pi_{t_{n_1}}(x_{t_{n_1}}^{(m,k)})}{q_{t_{n_1}}(x_{t_{n_1}}^{(m,k)})}$
7:    **end for**
8:    Normalize $\bar{w}_{t_{n_1}}^{(m,k)} = \dfrac{w_{t_{n_1}}^{(m,k)}}{\sum_{j=1}^{K} w_{t_{n_1}}^{(m,j)}}$
9:    Set $\mathrm{Node}_{t_{n_1}}^{(m)} = \{(x_{t_{n_1}}^{(m,k)}, \bar{w}_{t_{n_1}}^{(m,k)})\}_{k=1}^{K}$
10:   Compute node weight $\lambda_{t_{n_1}}^{(m)} = \dfrac{\sum_{k=1}^{K} w_{t_{n_1}}^{(m,k)}}{K}$
11: **end for**
12: **Step 2: Sequential Importance Sampling with Nested Resampling**
13: **for** $n = n_1 + 1, \ldots, n_2$ **do**
14:   Compute $\mathrm{ESS}_{t_{n-1}}^{\mathrm{node}} = \dfrac{(\sum_{m=1}^{M} \lambda_{t_{n-1}}^{(m)})^2}{\sum_{m=1}^{M} (\lambda_{t_{n-1}}^{(m)})^2}$
15:   **if** $\mathrm{ESS}_{t_{n-1}}^{\mathrm{node}} < M_\tau$ **then**
16:      Resample $\{\mathrm{Node}_{t_{n-1}}^{(m)}\}_{m=1}^{M}$ with weights proportional to $\{\lambda_{t_{n-1}}^{(m)}\}_{m=1}^{M}$
17:      Set $\lambda_{t_{n-1}}^{(m)} = 1$ for all $m$
18:   **end if**
19:   **for** $m = 1, \ldots, M$ **do**
20:      Resample $\{x_{t_{n_1}:t_{n-1}}^{(m,k)}\}_{k=1}^{K}$ using $\{\bar{w}_{t_{n-1}}^{(m,k)}\}_{k=1}^{K}$ to obtain $\{x_{t_{n_1}:t_{n-1}}^{(m,*k)}\}_{k=1}^{K}$
21:      **for** $k = 1, \ldots, K$ **do**
22:         Sample $x_{t_n}^{(m,k)} \sim q_{t_n}(\cdot | x_{t_{n-1}}^{(m,*k)})$
23:         Set $x_{t_{n_1}:t_n}^{(m,k)} = (x_{t_{n_1}:t_{n-1}}^{(m,*k)}, x_{t_n}^{(m,k)})$
24:         Compute $w_{t_n}^{(m,k)} = \dfrac{\pi_{t_n}(x_{t_{n_1}:t_n}^{(m,k)})}{q_{t_n}(x_{t_n}^{(m,k)}|x_{t_{n-1}}^{(m,*k)})\pi_{t_{n-1}}(x_{t_{n_1}:t_{n-1}}^{(m,*k)})}$
25:      **end for**
26:      Normalize $\bar{w}_{t_n}^{(m,k)} = \dfrac{w_{t_n}^{(m,k)}}{\sum_{j=1}^{K} w_{t_n}^{(m,j)}}$
27:      Set $\mathrm{Node}_{t_n}^{(m)} = \{(x_{t_{n_1}:t_n}^{(m,k)}, \bar{w}_{t_n}^{(m,k)})\}_{k=1}^{K}$
28:      Compute $\lambda_{t_n}^{(m)} = \lambda_{t_{n-1}}^{(m)} \cdot \dfrac{\sum_{k=1}^{K} w_{t_n}^{(m,k)}}{K}$
29:   **end for**
30: **end for**
31: **Step 3: Final Propagation**
32: **for** $m = 1, \ldots, M$ **do**
33:   **for** $k = 1, \ldots, K$ **do**
34:      Solve $\mathrm{d}x_t = v_{\theta^*}(x_t, t)\mathrm{d}t$ from $t = t_{n_2}$ to $t = 1$ to obtain $x_1^{(m,k)}$
35:      Compute $w_1^{(m,k)} = \lambda_{t_{n_2}}^{(m)} \cdot \bar{w}_{t_{n_2}}^{(m,k)} \cdot \dfrac{\tilde{p}(y|x_1^{(m,k)})}{\tilde{p}(y|\hat{x}_1(x_{t_{n_2}}^{(m,k)}))}$
36:   **end for**
37: **end for**
38: **Step 4: Output**
39: Normalize $\bar{w}_1^{(m,k)} = \dfrac{w_1^{(m,k)}}{\sum_{i=1}^{M} \sum_{j=1}^{K} w_1^{(i,j)}}$
40: **return** Weighted samples $\{(x_1^{(m,k)}, \bar{w}_1^{(m,k)}) : m = 1, \ldots, M, \ k = 1, \ldots, K\}$

---

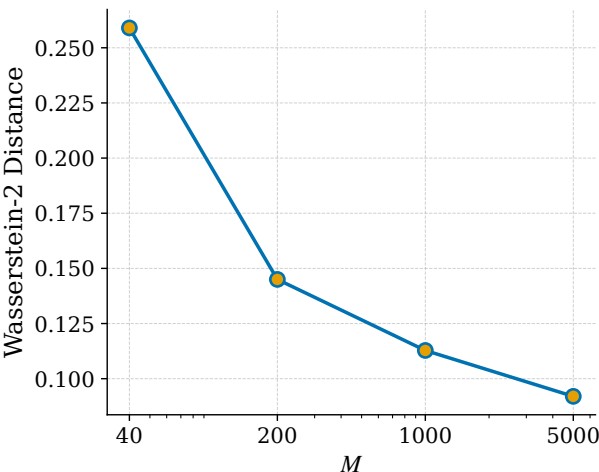

*Figure 6.* Wasserstein-2 distance versus number of nodes $M$ with $K = 4$ fixed.

### E.2. Proof of Proposition E.1

*Proof.* We begin by reorganizing $\sum_{m=1}^{M} \sum_{k=1}^{K} \bar{W}_1^{(m,k)} \phi(X_1^{(m,k)})$ as follows:

$$
\begin{aligned}
& \sum_{m=1}^{M} \sum_{k=1}^{K} \bar{W}_1^{(m,k)} \phi(X_1^{(m,k)}) \\
& = \frac{\sum_{m=1}^{M} \Lambda_{t_{n_2}}^{(m)} \left[ \sum_{k=1}^{K} \bar{W}_{t_{n_2}}^{(m,k)} \frac{\tilde{p}(y|X_1^{(m,k)})}{\tilde{p}(y|\hat{x}_1(X_{t_{n_2}}^{(m,k)}))} \phi(X_1^{(m,k)}) \right]}{\sum_{m=1}^{M} \Lambda_{t_{n_2}}^{(m)} \left[ \sum_{k=1}^{K} \bar{W}_{t_{n_2}}^{(m,k)} \frac{\tilde{p}(y|X_1^{(m,k)})}{\tilde{p}(y|\hat{x}_1(X_{t_{n_2}}^{(m,k)}))} \right]} \\
& = \frac{\sum_{m=1}^{M} \bar{\Lambda}_{t_{n_2}}^{(m)} \left[ \sum_{k=1}^{K} \bar{W}_{t_{n_2}}^{(m,k)} \frac{\tilde{p}(y|X_1^{(m,k)})}{\tilde{p}(y|\hat{x}_1(X_{t_{n_2}}^{(m,k)}))} \phi(X_1^{(m,k)}) \right]}{\sum_{m=1}^{M} \bar{\Lambda}_{t_{n_2}}^{(m)} \left[ \sum_{k=1}^{K} \bar{W}_{t_{n_2}}^{(m,k)} \frac{\tilde{p}(y|X_1^{(m,k)})}{\tilde{p}(y|\hat{x}_1(X_{t_{n_2}}^{(m,k)}))} \right]},
\end{aligned}
\tag{74}
$$

where $\bar{\Lambda}_{t_{n_2}}^{(m)} := \Lambda_{t_{n_2}}^{(m)} / \sum_{m'=1}^{M} \Lambda_{t_{n_2}}^{(m')}$ denotes the normalized node weight.

For each node $m$, the set of random variables generated during the intermediate stages consists of

$$
\left\{ X_{t_n}^{(m,k)} : n = n_1, \ldots, n_2, \, k = 1, \ldots, K \right\} \quad \text{and} \quad \left\{ X_{t_n}^{(m,*k)} : n = n_1, \ldots, n_2 - 1, \, k = 1, \ldots, K \right\},
$$

which arise from sampling from the proposal distributions and resampling. We view this set of random variables as a deterministic function of the random seed associated with node $m$, denoted by $u_{t_{n_1}:t_{n_2}}^{(m)}$. Specifically, $u_{t_{n_1}:t_{n_2}}^{(m)}$ serves as the base random variable that determines

$$
\left\{ X_{t_n}^{(m,k)} \right\}_{n,k} \quad \text{and} \quad \left\{ X_{t_n}^{(m,*k)} \right\}_{n,k}
$$

via the inverse CDF method applied to the proposal distributions together with the resampling rule and subsequently determines $\{W_{t_n}^{(m,k)} : n = n_1, \ldots, n_2, \, k = 1, \ldots, K\}$ and $\{X_1^{(m,k)} : k = 1, \ldots, K\}$. We denote the distribution of the random seed $U_{t_{n_1}:t_{n_2}}$ by $\prod_{n=n_1}^{n_2} \nu_{t_n}(u_{t_n}|u_{t_{n_1}:t_{n-1}})$. (In fact, it is not restrictive to assume that $\{U_{t_n}\}_{n=n_1}^{n_2}$ are independent; we adopt a more general notation here for flexibility.)

From this perspective, the outer-level reweighting and resampling over the $M$ nodes in Algorithm 2 can be interpreted as sequential importance sampling with proposal distribution $\{\nu_{t_n}(u_{t_n}|u_{t_{n_1}:t_{n-1}})\}_{n=n_1}^{n_2}$. The corresponding target distribution $\gamma_{t_n}(u_{t_{n_1}:t_n})$ is given by

$$
\gamma_{t_n}(u_{t_{n_1}:t_n}) = \frac{\hat{Z}_{t_n}(u_{t_{n_1}:t_n}) \prod_{i=n_1}^{n} \nu_{t_i}(u_{t_i}|u_{t_{n_1}:t_{i-1}})}{Z_{t_n}},
\tag{75}
$$

where

$$\hat{Z}_{t_n}(u_{t_{n_1}:t_n}) = \prod_{i=n_1}^{n} \frac{1}{K} \sum_{k=1}^{K} w_{t_i}^{(k)}, \tag{76}$$

and

$$Z_{t_n} = \int \hat{Z}_{t_n}(u_{t_{n_1}:t_n}) \prod_{i=n_1}^{n} \nu_{t_i}(u_{t_i}|u_{t_{n_1}:t_{i-1}}) \, \mathrm{d}u_{t_{n_1}:t_n}. \tag{77}$$

The incremental weight of a node at time $t = t_n$, corresponding to this proposal and target specification, is proportional to

$$\frac{\hat{Z}_{t_n}(u_{t_{n_1}:t_n}) \prod_{i=n_1}^{n} \nu_{t_i}(u_{t_i}|u_{t_{n_1}:t_{i-1}})}{\hat{Z}_{t_{n-1}}(u_{t_{n_1}:t_{n-1}}) \prod_{i=n_1}^{n-1} \nu_{t_i}(u_{t_i}|u_{t_{n_1}:t_{i-1}}) \cdot \nu_{t_n}(u_{t_n}|u_{t_{n_1}:t_{n-1}})}$$
$$= \frac{1}{K} \sum_{k=1}^{K} W_{t_n}^{(k)}, \tag{78}$$

which is exactly the node weight update rule specified in Algorithm 2.

Applying Del Moral (2004, Theorem 7.4.3), we have that

$$\sum_{m=1}^{M} \bar{\Lambda}_{t_{n_2}}^{(m)} \left[ \sum_{k=1}^{K} \bar{W}_{t_{n_2}}^{(m,k)} \frac{\tilde{p}(y|X_1^{(m,k)})}{\tilde{p}(y|\hat{x}_1(X_{t_{n_2}}^{(m,k)}))} \phi(X_1^{(m,k)}) \right] \xrightarrow{\text{a.s.}} \mathbb{E}_{\gamma_{t_{n_2}}} \left[ \sum_{k=1}^{K} \bar{W}_{t_{n_2}}^{(k)} \frac{\tilde{p}(y|X_1^{(k)})}{\tilde{p}(y|\hat{x}_1(X_{t_{n_2}}^{(k)}))} \phi(X_1^{(k)}) \right] \tag{79}$$

as $M \to \infty$. Similarly,

$$\sum_{m=1}^{M} \bar{\Lambda}_{t_{n_2}}^{(m)} \left[ \sum_{k=1}^{K} \bar{W}_{t_{n_2}}^{(m,k)} \frac{\tilde{p}(y|X_1^{(m,k)})}{\tilde{p}(y|\hat{x}_1(X_{t_{n_2}}^{(m,k)}))} \right] \xrightarrow{\text{a.s.}} \mathbb{E}_{\gamma_{t_{n_2}}} \left[ \sum_{k=1}^{K} \bar{W}_{t_{n_2}}^{(k)} \frac{\tilde{p}(y|X_1^{(k)})}{\tilde{p}(y|\hat{x}_1(X_{t_{n_2}}^{(k)}))} \right], \tag{80}$$

as $M \to \infty$.

Following Section D.1, we write $X_1 = T^*(X_{t_{n_2}})$. Applying Del Moral (2004, Theorem 7.4.2), we obtain

$$\mathbb{E}_{\gamma_{t_{n_2}}} \left[ \sum_{k=1}^{K} \bar{W}_{t_{n_2}}^{(k)} \frac{\tilde{p}(y|X_1^{(k)})}{\tilde{p}(y|\hat{x}_1(X_{t_{n_2}}^{(k)}))} \phi(X_1^{(k)}) \right]$$
$$= \frac{\int \hat{Z}_{t_{n_2}}(u_{t_{n_1}:t_{n_2}}) \left[ \sum_{k=1}^{K} \bar{w}_{t_{n_2}}^{(k)} \frac{\tilde{p}(y|T^*(x_{t_{n_2}}^{(k)}))}{\tilde{p}(y|\hat{x}_1(x_{t_{n_2}}^{(k)}))} \phi(T^*(x_{t_{n_2}}^{(k)})) \right] \prod_{n=n_1}^{n_2} \nu_{t_n}(u_{t_n}|u_{t_{n_1}:t_{n-1}}) du_{t_{n_1}:t_{n_2}}}{Z_{t_{n_2}}} \tag{81}$$
$$= \frac{\int \frac{\tilde{p}(y|T^*(x_{t_{n_2}}))}{\tilde{p}(y|\hat{x}_1(x_{t_{n_2}}))} \phi(T^*(x_{t_{n_2}})) \pi_{t_{n_2}}(x_{t_{n_1}:t_{n_2}}) dx_{t_{n_1}:t_{n_2}}}{Z_{t_{n_2}}},$$

and

$$\mathbb{E}_{\gamma_{t_{n_2}}} \left[ \sum_{k=1}^{K} \bar{W}_{t_{n_2}}^{(k)} \frac{\tilde{p}(y|X_1^{(k)})}{\tilde{p}(y|\hat{x}_1(X_{t_{n_2}}^{(k)}))} \right]$$
$$= \frac{\int \hat{Z}_{t_{n_2}}(u_{t_{n_1}:t_{n_2}}) \left[ \sum_{k=1}^{K} \bar{w}_{t_{n_2}}^{(k)} \frac{\tilde{p}(y|T^*(x_{t_{n_2}}^{(k)}))}{\tilde{p}(y|\hat{x}_1(x_{t_{n_2}}^{(k)}))} \right] \prod_{n=n_1}^{n_2} \nu_{t_n}(u_{t_n}|u_{t_{n_1}:t_{n-1}}) du_{t_{n_1}:t_{n_2}}}{Z_{t_{n_2}}} \tag{82}$$
$$= \frac{\int \frac{\tilde{p}(y|T^*(x_{t_{n_2}}))}{\tilde{p}(y|\hat{x}_1(x_{t_{n_2}}))} \pi_{t_{n_2}}(x_{t_{n_1}:t_{n_2}}) dx_{t_{n_1}:t_{n_2}}}{Z_{t_{n_2}}}.$$

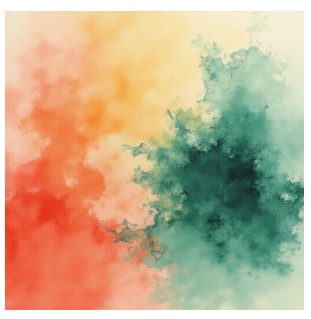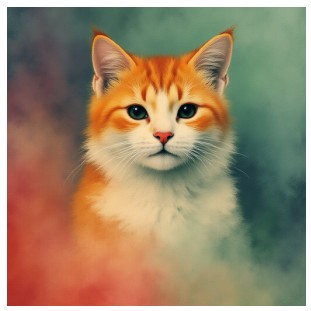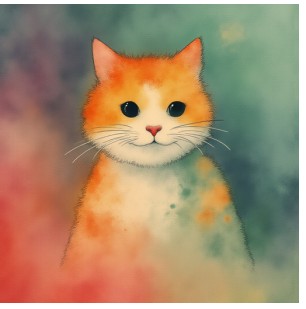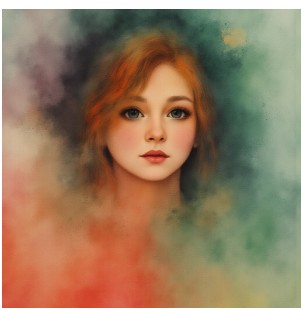

*Figure 7.* Qualitative results of style-guided generation using the FLUX model. The *leftmost* image is the style reference; the remaining three are generated conditioned on it.

Following exactly the same argument as in Equations (56) and (57), we have

$$
\frac{\int \frac{\tilde{p}(y|T^*(x_{t_{n_2}}))}{\tilde{p}(y|\hat{x}_1(x_{t_{n_2}}))} \phi(T^*(x_{t_{n_2}})) \pi_{t_{n_2}}(x_{t_{n_1}:t_{n_2}}) dx_{t_{n_1}:t_{n_2}}}{\int \frac{\tilde{p}(y|T^*(x_{t_{n_2}}))}{\tilde{p}(y|\hat{x}_1(x_{t_{n_2}}))} \pi_{t_{n_2}}(x_{t_{n_1}:t_{n_2}}) dx_{t_{n_1}:t_{n_2}}} = \frac{\int \frac{\tilde{p}(y|T^*(x_{t_{n_2}}))}{\tilde{p}(y|\hat{x}_1(x_{t_{n_2}}))} \phi(T^*(x_{t_{n_2}})) \pi^*_{t_{n_2}}(x_{t_{n_2}}) dx_{t_{n_2}}}{\int \frac{\tilde{p}(y|T^*(x_{t_{n_2}}))}{\tilde{p}(y|\hat{x}_1(x_{t_{n_2}}))} \pi^*_{t_{n_2}}(x_{t_{n_2}}) dx_{t_{n_2}}} = \mathbb{E}_{p_{\text{data}}(\cdot|y)}[\phi].
$$

(83)

where $\pi^*_{t_{n_2}}$ denotes the normalized marginal density of $x_{t_{n_2}}$ under the intermediate target distribution $\pi_{t_{n_2}}(x_{t_{n_1}:t_{n_2}})$.

In conclusion, we have demonstrated that

$$
\sum_{m=1}^{M}\sum_{k=1}^{K} \bar{W}_1^{(m,k)} \phi(X_1^{(m,k)}) \xrightarrow{\text{a.s.}} \mathbb{E}_{p_{\text{data}}(\cdot|y)}[\phi]
$$

(84)

as $M \to \infty$. $\qquad\qquad\square$

# F. Additional Experimental Details

### F.1. Toy Example

**Sampling:** We use $N = 1000$ integration steps with Euler (for ODE) and Euler–Maruyama (for SDE) discretization, corresponding to a step size of $\Delta t = 0.001$.

**TFTF:** $\alpha(t) = 1/t$, $\beta(t) = 1$, $[t_{n_1}, t_{n_2}] = [0.4, 0.8]$. In practice, we employ systematic resampling rather than multinomial resampling due to its superior computational efficiency.

**FMPS (Song et al., 2025):** $r = 1$.

**D-Flow (Ben-Hamu et al., 2024):** Adam optimizer with learning rate 0.1, 20 optimization iterations, MSE cost function.

**FlowChef (Patel et al., 2025):** $s' = \Delta t$, cost function from Equation (3) of Patel et al. (2025).

All experiments were conducted on an NVIDIA RTX 3090 GPU.

### F.2. MNIST

**Unconditional Model:** U-Net trained from scratch with batch size 128, Adam optimizer (learning rate $2 \times 10^{-4}$), for 100 epochs. Images are converted from PIL format to tensors and normalized to $[-1, 1]$.

**Likelihood-Specifying Classifier:** ResNet-50 (He et al., 2016) from `https://github.com/VSehwag/minimal-diffusion`, consistent with TDS (Wu et al., 2023).

**External Classifier:** Super Learner model from `https://github.com/Curt-Park/handwritten_digit_recognition`.

**Sampling:** $N = 800$ steps, $\Delta t = 0.00125$.

**TFTF:** $\alpha(t) = 4/t$ for Table 1; $\alpha(t) \in \{1/t, 2/t, 4/t, 8/t\}$ for ablation. $\beta(t) = 1$, $[t_{n_1}, t_{n_2}] = [3/16, 1/2]$.

**FMPS:** $r = 0.1$.

**FlowChef:** $s' = 2\Delta t$, cost function from Equation (5) of Patel et al. (2025).

Baseline hyperparameters were determined via grid search. Experiments were conducted on NVIDIA RTX 3090 and RTX 4090 GPUs.

## F.3. CIFAR-10

**Unconditional Model:** Pretrained rectified flow model (Liu et al., 2022a) from `https://github.com/gnobitab/RectifiedFlow`.

**Likelihood-Specifying Classifier:** VGG-13-BN (Simonyan & Zisserman, 2014) from `https://github.com/huyvnphan/PyTorch_CIFAR10`.

**External Classifier:** DenseNet-121 (Huang et al., 2017) from `https://github.com/huyvnphan/PyTorch_CIFAR10`.

**Evaluation Metrics:** Inception Score (IS) (Salimans et al., 2016) uses Inception-v3 (Szegedy et al., 2016) from `torchvision.models.inception_v3` with 10 splits. For intra-class Wasserstein-2 distance, we extract 2048-dimensional features using Inception-v3 and compute the feature-space $\mathcal{W}_2$ distance between generated samples and dataset images of the corresponding class.

**Sampling:** $N = 800$ steps, $\Delta t = 0.00125$.

**TFTF:** $\alpha(t) = 4/t$, $\beta(t) = 1$. For Table 2, $[t_{n_1}, t_{n_2}] = [7/16, 14/16]$. For sensitivity analysis, we vary $t_{n_1} \in \{5/16, 6/16, 7/16, 8/16, 9/16\}$ with $t_{n_2} = 14/16$ fixed, and $t_{n_2} \in \{10/16, 11/16, 12/16, 13/16, 14/16\}$ with $t_{n_1} = 7/16$ fixed. Outer-level resampling threshold $M_\tau = 800$ for Nested TFTF.

**FMPS:** $r = 0.025$.

**FlowChef:** $s' = 5\Delta t$, cost function from Equation (5) of Patel et al. (2025).

Baseline hyperparameters were determined via grid search. Experiments were conducted on NVIDIA RTX 3090 and RTX 4090 GPUs.

## F.4. CelebA-HQ

**Unconditional Model:** Pretrained rectified flow model (Liu et al., 2022a) from `https://github.com/gnobitab/RectifiedFlow`.

**Text Prompts:** Constructed based on attribute combinations from `https://github.com/BillyXYB/TransEditor`. Each prompt consists of one positive text and multiple negative texts (provided in Supplementary Material).

**Likelihood-Specifying Model:** ConvNext-XXLarge CLIP (Radford et al., 2021; Liu et al., 2022b; Cherti et al., 2023) from `https://github.com/mlfoundations/open_clip`. Likelihood is computed as the softmax (temperature $\tau = 0.01$) of cosine similarity between image features and positive text, against the maximum cosine similarity with negative texts.

**External Classifier:** Attribute classifiers from `https://github.com/BillyXYB/TransEditor`. Due to defects in the blonde hair classifier, we use human evaluation for this attribute. All-attributes accuracy ($\text{ACC}_E$) represents the frequency at which all specified attributes are satisfied.

**Sampling:** $N = 400$ steps, $\Delta t = 0.0025$.

**TFTF:** $\alpha(t) = 2/t$, $\beta(t) = 1$, $[t_{n_1}, t_{n_2}] = [3/40, 2/5]$, post-resampling diversification interval length $\delta = 1/10$.

**FMPS:** $r = 0.003$.

**FlowChef:** $s' = 0.1$, cost function from Equation (5) of Patel et al. (2025).

*Table 5.* Ablation study comparing the original and accelerated versions on CIFAR-10 class-conditional sampling. $\text{ACC}_L$: likelihood-specifying classifier accuracy; $\text{ACC}_E$: external classifier accuracy; IS: Inception Score; $\mathcal{W}_2$: intra-class feature-space Wasserstein-2 distance (averaged across classes); ESS: effective sample size (averaged across classes).

| Method | $\text{ACC}_L$ | $\text{ACC}_E \uparrow$ | IS $\uparrow$ | $\mathcal{W}_2 \downarrow$ | ESS $\uparrow$ |
|---|---|---|---|---|---|
| Nested TFTF | 92.56% | 92.82% | **8.88** $_{\pm 0.08}$ | 32.61 | 13000.80/16000 |
| Nested TFTF (Accelerated) | 92.47% | **92.94%** | 8.60 $_{\pm 0.06}$ | **32.33** | **13428.10/16000** |

Baseline hyperparameters were determined via grid search. Experiments were conducted on NVIDIA RTX 4090 and RTX 5090 GPUs.

### F.5. Evaluation Protocol for TFTF

Since Algorithms 1 and 2 produce *weighted* samples $\{(x_1^{(k)}, \bar{w}_1^{(k)})\}_{k=1}^K$, we adopt the following evaluation protocol to ensure that all metrics properly account for the sample weights.

**Weighted-average metrics.** The likelihood-specifying classifier accuracy ($\text{ACC}_L$), external classifier accuracy ($\text{ACC}_E$), and CLIP score ($\text{CLIP}_L$) are computed as weighted averages over the particle set.

**Resampling-based metrics.** Metrics whose standard evaluation procedures assume equally weighted samples—namely, intra-batch pairwise $\ell_2$ distance, Inception Score (IS), and intra-class feature-space Wasserstein-2 distance ($\mathcal{W}_2$)—are evaluated as follows. We first draw an equally weighted sample set by resampling from $\{(x_1^{(k)}, \bar{w}_1^{(k)})\}_{k=1}^K$ with probabilities proportional to $\{\bar{w}_1^{(k)}\}_{k=1}^K$, and then apply the standard evaluation procedure to the resulting unweighted samples.

## G. Additional Experiments

### G.1. Accelerated Variant and Computational Complexity Analysis

As stated in Section 3.1.1, we construct the conditional velocity field $v_c$ by first projecting the noise-corrupted intermediate state $x(t)$ onto a predicted final state $\hat{x}_1(x(t))$ via a single Euler step:

$$\hat{x}_1(x(t)) = x(t) + (1 - t) \cdot v_{\theta^*}(x(t), t), \tag{85}$$

and then defining the practical conditional velocity field as

$$v_c(x(t), t) = v_{\theta^*}(x(t), t) + \frac{1 - t}{t} \cdot \beta(t) \cdot \nabla_{x(t)} \log \tilde{p}(y|\hat{x}_1(x(t))). \tag{86}$$

Applying the chain rule, the gradient term $\nabla_{x(t)} \log \tilde{p}(y|\hat{x}_1(x(t)))$ in Equation (86) can be expanded as

$$\nabla_{x(t)} \log \tilde{p}(y|\hat{x}_1(x(t))) = \left[ I + (1 - t) \frac{\partial v_{\theta^*}(x(t), t)}{\partial x(t)} \right]^\top \nabla_{\hat{x}_1(x(t))} \log \tilde{p}(y|\hat{x}_1(x(t))). \tag{87}$$

The primary computational burden stems from the Jacobian term $\frac{\partial v_{\theta^*}(x(t),t)}{\partial x(t)}$, which requires backpropagation through the pretrained velocity field. To reduce this cost, we propose omitting the term $(1 - t) \frac{\partial v_{\theta^*}(x(t),t)}{\partial x(t)}$ and directly approximating $\nabla_{x(t)} \log \tilde{p}(y|\hat{x}_1(x(t)))$ with $\nabla_{\hat{x}_1(x(t))} \log \tilde{p}(y|\hat{x}_1(x(t)))$. Notably, this simplification preserves the asymptotic accuracy of Algorithms 1 and 2: since $v_c$ is used solely for constructing proposal distributions, the proofs in Sections D.1 and E.2 remain valid for this accelerated variant.

To empirically validate this acceleration technique, we revisit the class-conditional sampling task on CIFAR-10 from Section 4.2, applying it to Nested TFTF (Algorithm 2) while keeping all other hyperparameters unchanged. As shown in Table 5, the original and accelerated versions achieve comparable classification accuracy, Inception Score, $\mathcal{W}_2$ distance, and ESS. A detailed computational complexity analysis of various methods is provided in Table 6.

*Table 6.* Computational complexity comparison of different methods. $N$: number of Euler (or Euler–Maruyama) steps; $K$: number of particles; $M$: number of optimization iterations for the initial point; $[t_{n_1}, t_{n_2}]$: resampling interval where $0 < t_{n_1} < t_{n_2} < 1$.

| Method | Forward Eval. | Backward Prop. | Likelihood Grad. | Wall-Clock Time[†] |
|---|---|---|---|---|
| D-Flow | $(M+1)NK$ | $MNK$ | $MK$ | $-^a$ |
| TDS | $NK$ | $NK$ | $NK$ | 2min 30s[b] |
| FMPS | $NK$ | $NK$ | $NK$ | 1min 46s |
| FlowChef | $NK$ | $0$ | $NK$ | 1min 07s |
| TFTF | $NK$ | $NK \cdot (t_{n_2} - t_{n_1})$ | $NK \cdot (t_{n_2} - t_{n_1})$ | 1min 05s[c] |
| TFTF (Accelerated) | $NK$ | $0$ | $NK \cdot (t_{n_2} - t_{n_1})$ | 46s[c] |

[†]Measured on a single NVIDIA RTX 3090 for CIFAR-10 class-conditional generation with $N = 800$ and $K = 16$.
[a]D-Flow's wall-clock time depends on the number of optimization iterations $M$.
[b]Assuming the denoising network and velocity-field network have the same architectural complexity.
[c]With $[t_{n_1}, t_{n_2}] = [7/16, 12/16]$.

### G.2. Sensitivity Analysis on the Resampling Interval

In this section, we investigate the sensitivity of Algorithm 1 to the choice of resampling interval $[t_{n_1}, t_{n_2}]$ on CIFAR-10, using the accelerated variant proposed in Section G.1. Throughout the experiments, we fix the following hyperparameters: particle count $K = 25$, guidance scale $\beta(t) = 1$, and $\alpha(t) = 4/t$.

Figure 10 illustrates the variation in $\text{ACC}_E$ across different target classes for various values of $t_{n_1}$, with $t_{n_2} = 14/16$ held fixed. Figure 11 presents the corresponding results for varying $t_{n_2}$ values while maintaining $t_{n_1} = 7/16$ constant. As shown, Algorithm 1 exhibits robust performance across different choices of the intermediate interval $[t_{n_1}, t_{n_2}]$, with the mean $\text{ACC}_E$ averaged over all classes consistently exceeding 90% for all configurations examined.

Furthermore, the results in Figure 10 reveal that initiating resampling earlier does not necessarily yield improved performance. While smaller values of $t_{n_1}$ advance the timing of particle pruning and enable promising particles to amplify themselves at earlier stages, samples in the early phase of the generation process tend to be noisy, and thus the look-ahead function may fail to provide sufficient information about the final state. This inherent trade-off accounts for the observed decline in $\text{ACC}_E$ for the "cat" class at $t_{n_1} = 8/16$ and for the "frog" class at $t_{n_1} = 6/16$ in Figure 10. The discrepancy in the timing of these drops further suggests that the emergence of class-discriminative features occurs asynchronously across different classes. Additionally, the results presented in Figure 11 demonstrate that moderately early termination of the resampling procedure can reduce computational overhead without compromising classification accuracy, while avoiding the loss of particle diversity that may result from late-stage resampling.

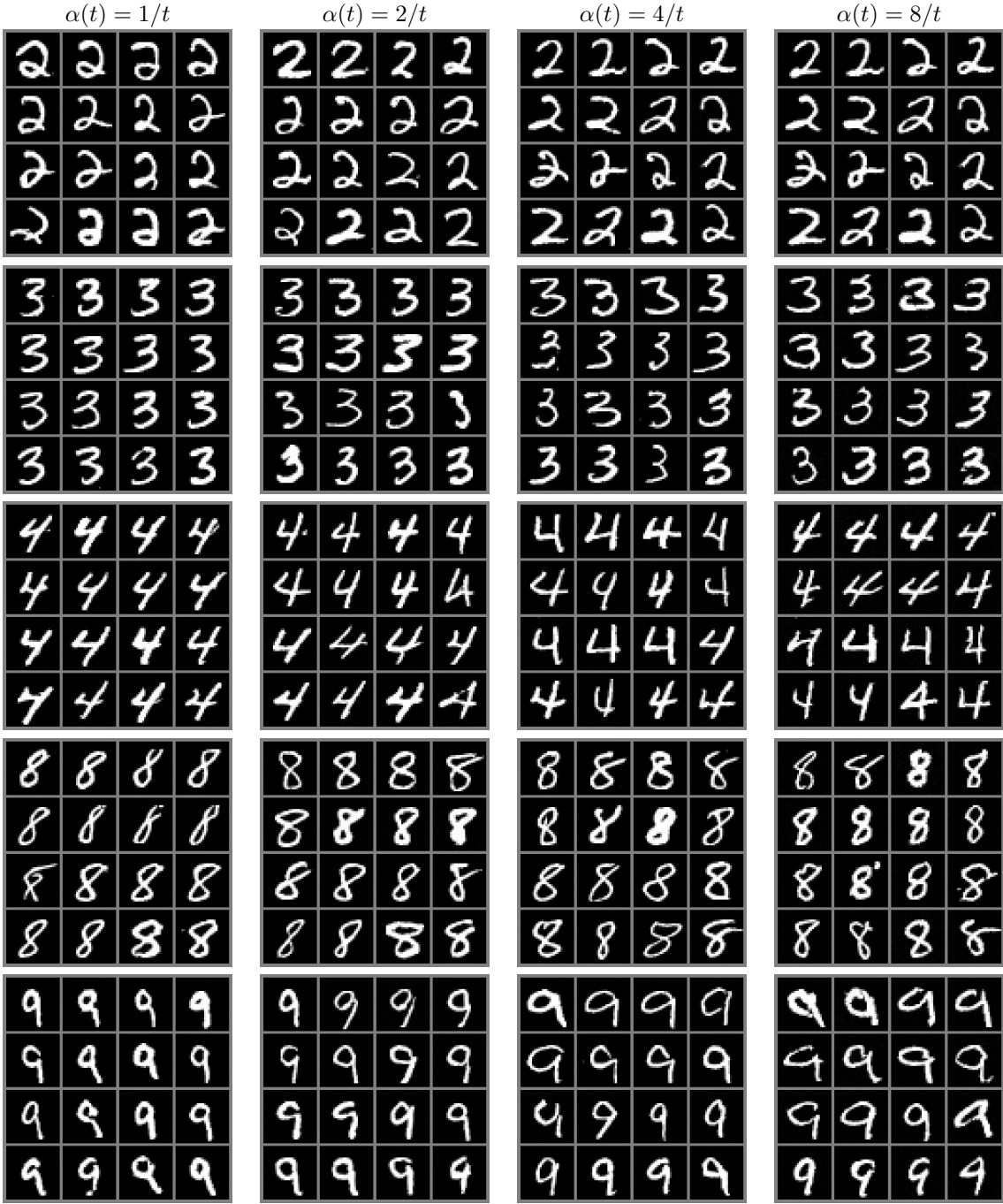

*Figure 8.* Generated samples on MNIST using TFTF (Algorithm 1) with $K = 16$ under different $\alpha(t)$ configurations. Each column corresponds to a different $\alpha(t) \in \{1/t, 2/t, 4/t, 8/t\}$, while each row shares the same random seed.

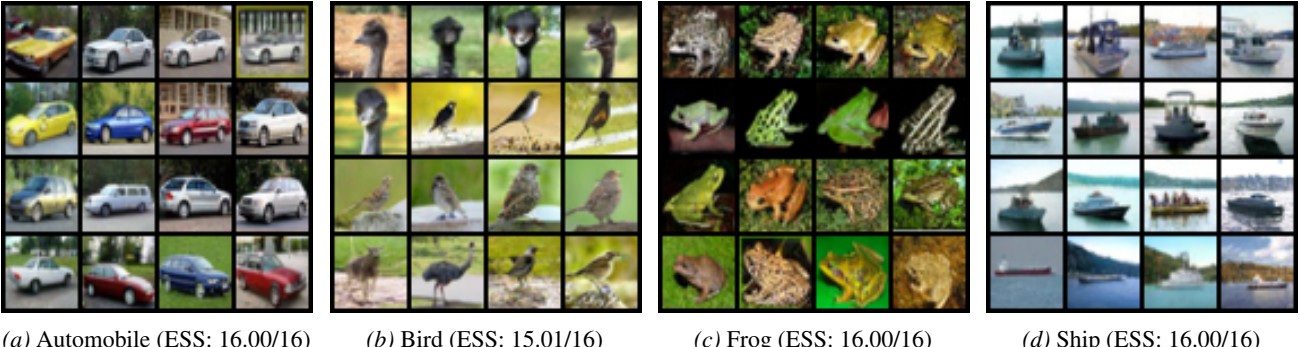

*(a)* Automobile (ESS: 16.00/16)     *(b)* Bird (ESS: 15.01/16)     *(c)* Frog (ESS: 16.00/16)     *(d)* Ship (ESS: 16.00/16)

*Figure 9.* Generated samples on CIFAR-10 using TFTF (Algorithm 1) with $K = 16$.

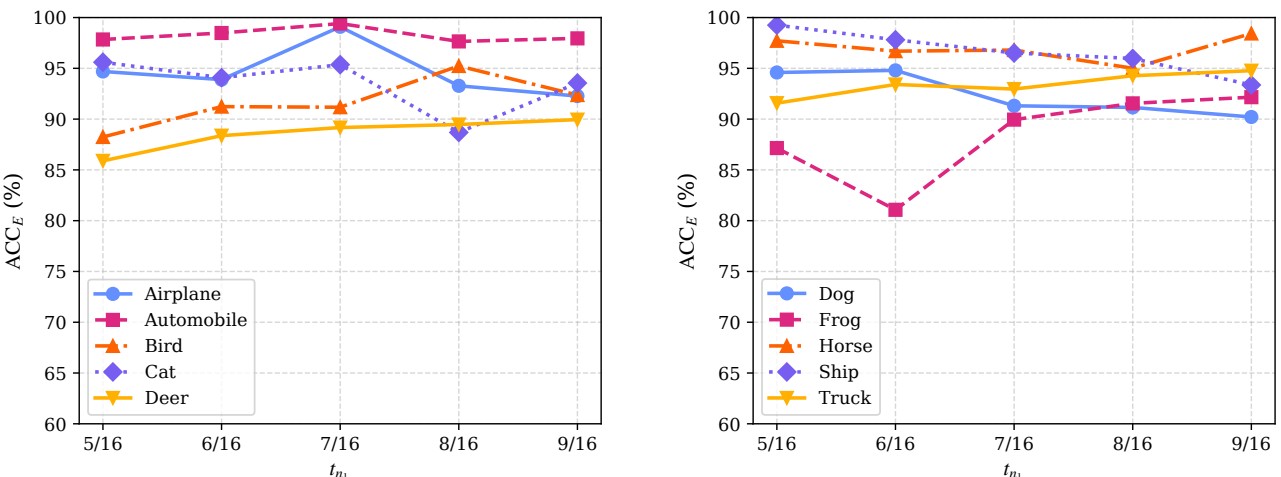

*Figure 10.* Classification accuracy across different target classes as a function of $t_{n_1}$, with $t_{n_2} = 14/16$ held fixed.

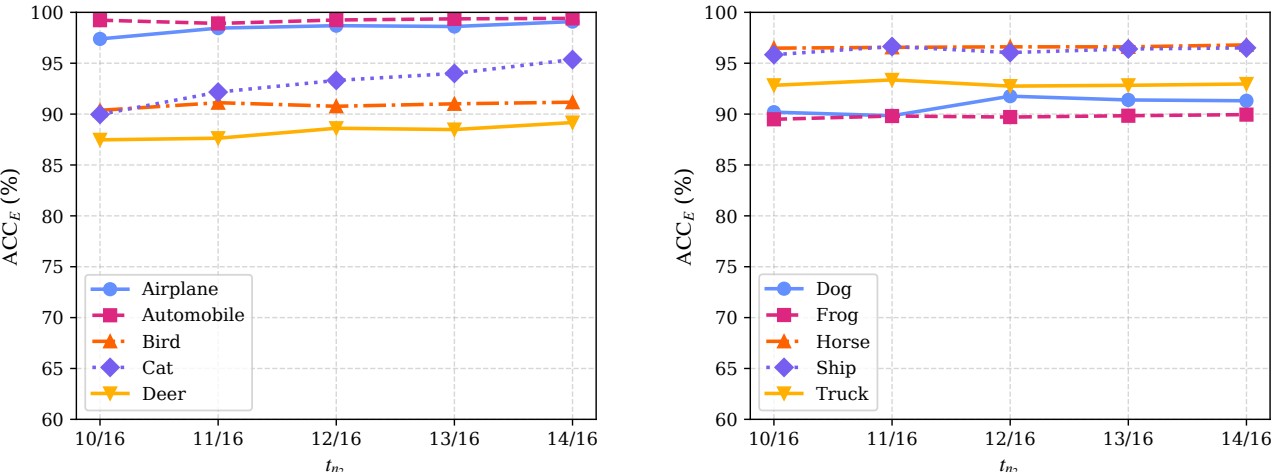

*Figure 11.* Classification accuracy across different target classes as a function of $t_{n_2}$, with $t_{n_1} = 7/16$ held fixed.

