# OpenReview forum: "TFTF: Training-Free Targeted Flow for Conditional Sampling"
_ICML.cc/2026/Conference — ICML 2026 regular_

### Official Review · Reviewer_AesQ · 2026-03-08

**Soundness:** 3
**Presentation:** 3
**Significance:** 3
**Originality:** 4
**Overall Recommendation:** 5
**Confidence:** 3

**Summary:**

TFTF proposes a **training-free** method for **conditional sampling** with flow matching / rectified-flow generators. Given a pretrained **unconditional** flow model that samples from $p_{\text{data}}(x)$ and an **off-the-shelf scoring / likelihood model** $\tilde p(y\mid x)$ (e.g., a classifier for class labels or CLIP for text), the goal is to generate samples from the conditional distribution $p_{\text{data}}(x\mid y)\propto p_{\text{data}}(x)\tilde p(y\mid x)$ without fine-tuning the generator. The method builds a guided sampling procedure that steers samples toward satisfying $y$, then corrects the remaining mismatch using a principled **reweighting-and-selection** scheme over multiple parallel candidates during an intermediate time window. To make selection meaningful, TFTF introduces an **intermediate stochastic (SDE) phase** between deterministic ODE segments so duplicated candidates can evolve into different outcomes. A final correction step yields weighted samples with an **asymptotic accuracy** guarantee as the number of candidates increases. Experiments on a toy problem, MNIST/CIFAR-10 class conditioning, and CelebA-HQ text-to-image demonstrate improved conditional adherence and competitive quality versus existing training-free baselines.

**Compliance With Llm Reviewing Policy:**

Affirmed.

**Final Justification:**

All my concerns have been addressed.

**Key Questions For Authors:**

This paper is not easy to read, many notations are not defined clearly. It would be help, if the retionale of the notation is stated.

**Limitations:**

yes

**Strengths And Weaknesses:**

## Soundness

**Strengths**

* The core technical decomposition of the *optimal conditional* velocity field (Eq. 8) is clearly stated and the proof is also provided in A.2.
* The method includes implementation safeguards to satisfy boundedness conditions (adding $\epsilon_1,\epsilon_2$, bounding $\alpha,\beta$, gradient clipping), and explicitly argues these do not change asymptotic accuracy.

**Weaknesses / assumptions that should be stated more explicitly**

* The $(1-t)/$ factor is singular as $t\to 0$; the paper defines the boundary by a limit, but stability/behavior near $t=0$ could be discussed more carefully.

## Presentation

**Strengths**

* Clear high-level pipeline: derive an ideal conditional field, introduce an implementable surrogate, then correct with weighting/resampling (Algorithm 1).
* Experiments include diagnostic evidence for the intended effect of resampling (ESS improvement and qualitative results).

**Weaknesses / suggestions**

* Notation is heavy $\pi,q,h,f$, multiple time windows $[t_{n_1},t_{n_2}]$, $\tilde{X}, X$, $\tilde{p}, p$ and it’s easy to miss that $\pi_{t_n}$ is an *unnormalized path-space* target used only through ratios. A short “what is computed in practice” box (weights in incremental form) would help.

## Significance

**Strengths**

* Practical problem: conditional generation without retraining the generator is valuable when conditions change often or training is expensive.
* The paper addresses known weaknesses of purely heuristic steering by adding a principled correction mechanism (sample weighting + resampling) and provides an asymptotic-accuracy narrative.

**Weaknesses**

* The method adds inference-time compute (multiple particles, repeated likelihood evaluations) and has hyperparameters (e.g., $\alpha(t)$, resampling schedule). It would help to report compute/quality tradeoffs more explicitly.
* “Diversity/mode coverage” remains delicate: the paper itself notes late-stage resampling may reduce diversity and proposes early termination as a remedy.

## Originality

**Strengths**

* The main novelty is a thoughtful combination of (i) flow-matching conditional guidance, (ii) an SMC-style correction framework, and (iii) an ODE–SDE–ODE strategy to make resampling meaningful in otherwise deterministic FM dynamics.
* The paper positions itself clearly against optimization-based training-free methods and purely surrogate-velocity guidance, arguing weighting/resampling addresses bias/seed sensitivity and improves ESS.

---

> ### Author Rebuttal · Authors · 2026-03-31
>
> We sincerely thank the reviewer for providing us with many constructive comments. All concerns are addressed, and a point-by-point response is given below.
>
> ---
>
> ## 1. Behavior of the $\frac{1-t}{t}$ Factor near $t = 0$
>
> Thank you for this careful observation. We provide both an analytical discussion and a practical justification.
>
> **Analytical example.** Suppose $p_{\text{data}} = \mathcal{N}(\mu_1, \sigma_1^2)$ and $Y | \tilde{X}(1) = x_1 \sim \mathcal{N}(a x_1 + b\, \sigma_2^2)$. In this case, we can derive in closed form that
>
> $$\nabla_{x_t} \log p_{Y|\tilde{X}(t)}(y \mid x_t) = \frac{\alpha_t}{\sigma_{y|t}} \bigl(y - \alpha_t x_t - \beta_t\bigr),$$
>
> where
>
> $$\alpha_t = \frac{t \ a \ \sigma_1^2}{\sigma_t^2}, \quad \beta_t = b + \frac{a \ \mu_1 \ (1-t)^2}{\sigma_t^2}, \quad \sigma_{y|t} = \sigma_2^2 + \frac{a^2 \sigma_1^2 (1-t)^2}{\sigma_t^2}, \quad \sigma_t^2 = t^2 \sigma_1^2 + (1-t)^2.$$
>
> Therefore, the product in Eq. (8) satisfies
>
> $$\lim_{t \to 0^+} \frac{1-t}{t} \ \nabla_{x_t} \log p_{Y|\tilde{X}(t)}(y \mid x_t) = \frac{a \sigma_1^2}{\sigma_2^2 + a^2 \sigma_1^2} \bigl(y - a \mu_1 - b\bigr),$$
> which is finite. That is, the $\frac{1}{t}$ singularity is exactly cancelled by the vanishing of $\nabla_{x_t} \log p_{Y|\tilde{X}(t)}$ at rate $O(t)$.
>
> **In practice,** although the likelihood may be complicated, TFTF **completely avoids the singularity issue by design**: we only compute the guidance term during $[t_{n_1}, t_{n_2}]$ where $0 < t_{n_1} < t_{n_2} < 1$, so $\frac{1-t}{t}$ is never evaluated near $t = 0$.
>
> ---
>
> ## 2. Notation System
> We appreciate the suggestion to clarify our notation. To summarize the key conventions:
>
> - **SMC framework:** $\pi$ denotes the target; $q$ denotes the proposal induced by replacing $v_{\theta^*}$ with $v_c$ in SDE (12); $h$ denotes the transition density of SDE (12); $f$ denotes the marginal density of SDE (12).
> - **Tilde vs. plain notation:** $\tilde{X}$ denotes the stochastic process defined in Sec 2.1 and $\tilde{p}$ denotes the likelihood. $X$ and $p$ denote general processes and densities, respectively.
> - **Ratio trick:** As the reviewer correctly noted, $\pi_{t_n}$ is an unnormalized path-space target, and through the ratio-based weight computation, all normalizing constants cancel, and we **only need to know $\pi$ up to a constant**.
>
> In the final version, we will add a notation table summarizing all symbols and their relationships, include a **"What Is Computed in Practice" box**, and state the ratio trick more explicitly.
>
> ---
>
> ## 3. Compute/Quality Tradeoffs
>
> Thank you for this important point. In the appendix we already provide a complexity analysis and wall-clock time comparisons on CIFAR-10 in Table 5 (where TFTF achieves the shortest wall-clock time at matched sample count). We now supplement this with a **detailed comparison on MNIST** under $K = 4$:
>
> | Method | Wall-Clock Time | Velocity Evals | Velocity Backprops | Likelihood Backprops | Mean $ACC_E$ | Best-of-$K$ $ACC_E$ |
> |:---|:---:|:---:|:---:|:---:|:---:|:---:|
> | TFTF | 18s | 3200 | 1000 | 1000 | **95.28%** | — |
> | TFTF (Accelerated) | **11s** | 3200 | 0 | 1000 | **92.99%** | — |
> | FMPS | 40s | 3200 | 3200 | 3200 | 39.24% | 83.73% |
> | FlowChef | 12s | 3200 | 0 | 3200 | 23.57% | 64.35% |
>
> (measured on a single NVIDIA RTX 4090)
>
> 1. **TFTF (Accelerated) is the fastest** because it only computes the guidance term during $[t_{n_1}, t_{n_2}]$ and  avoids backpropagation through the velocity network entirely. The non-accelerated TFTF is between FlowChef and FMPS in speed, yet achieves dramatically better $ACC_E$.
>
> 2. **Mean $ACC_E$ vs. Best-of-$K$ $ACC_E$.** Since baseline methods' $ACC_E$ is independent of $K$, we also report their *best-of-$K$*  $ACC_E$ (i.e., selecting the sample with the highest likelihood) for a stricter comparison. Even under this favorable metric, baselines (83.73% and 64.35%) fall short of TFTF (95.28%). We emphasize that a high mean  $ACC_E$ is more valuable than a high best-of-$K$  $ACC_E$: TFTF produces $K$ diverse, condition-compliant samples, whereas best-of-$K$ selection only yields a single satisfactory sample out of $K$ candidates.
>
> 3. **Distributional guarantees.** TFTF's $K$ weighted particles converge to $p_{\text{data}}(\cdot \mid y)$ as $K \to \infty$. In contrast, for baselines, no matter how many particles are used, the sampling distribution retains a systematic gap from the target.
>
> We further provide a detailed ablation over $K$ on MNIST (see [**here (PDF)**](https://anonymous.4open.science/r/2026-814E/output_computational_analysis.pdf)). As shown, TFTF's ACC$_\text{E}$ **already surpasses all baselines with $K=1$**. Although total NFE scales linearly with $K$, the $K$ particles can be propagated **in parallel**, so wall-clock time remains essentially constant as $K$ grows. We also refer the reviewer to the **resampling schedule ablations** (Figures 9 and 10 in the appendix), which demonstrate TFTF's **robustness to hyperparameter choices**.

---

> > ### Author Rebuttal · Reviewer_AesQ · 2026-04-03
> >
> > All my concerns have been addressed.

---

> > > ### Author Response · Authors · 2026-04-08
> > >
> > > We are grateful for the time and thoughtfulness you invested in reviewing our work. It is encouraging to know that our responses were able to resolve all of your concerns.

---

### Official Review · Reviewer_omqp · 2026-03-11

**Soundness:** 2
**Presentation:** 3
**Significance:** 2
**Originality:** 3
**Overall Recommendation:** 5
**Confidence:** 3

**Summary:**

This paper adopts importance sampling for training-free conditional sampling. The authors incorporate a sequential Monte Carlo during intermediate stages of the generation process. They verify the effectiveness of the method on the MNIST, CIFAR-10, and CelebA-HQ datasets.

**Compliance With Llm Reviewing Policy:**

Affirmed.

**Final Justification:**

The strengths of this paper mainly lie in originality and soundness. During the rebuttal, the authors provide some necessary comparison with a series of works, which somehow improves the significance.

**Key Questions For Authors:**

The conditional generation is a special case of guidance. I am wondering how Proposition 3.1 is equivalent to Theorem 3.1 in [1].

[1]: Feng, R., Wu, T., Yu, C., Deng, W., & Hu, P. On the Guidance of Flow Matching. ICML 2025.

**Strengths And Weaknesses:**

Strengths:

1. The formulation of classifier guidance for flow matching is novel.

2. The mechanism of resampling in various sampling procedures is novel and reasonable.

Weaknesses:

Conditional generation, classifier guidance, and training-free guidance have been studied by many works. In general, conditional generation is a special case of guidance [1] and posterior sampling[2,3]. The papers lack sufficient comparison in the related work and experimental results.

[1]: Feng, R., Wu, T., Yu, C., Deng, W., & Hu, P. On the Guidance of Flow Matching. ICML 2025.

[2]: Wang, L., Cheng, C., Liao, Y., Qu, Y., & Liu, G. Training Free Guided Flow Matching with Optimal Control. ICLR 2025.

[3]: Song, K., Lai, H., Pan, Y., Yue, K., & Yin, J. (2024). Flow Matching Posterior Sampling: A Training-free Conditional Generation for Flow Matching.

---

> ### Author Rebuttal · Authors · 2026-03-31
>
> We sincerely thank the reviewer for providing us with many constructive comments. All concerns are addressed, and a point-by-point response is given below.
>
> ---
>
> ## 1. Mathematical Equivalence of Conditional Generation and Guidance
>
> We are grateful to the reviewer for pointing out connections of TFTF with energy guidance and posterior sampling [1,2,3]. After examining these works carefully, we are happy to report that our TFTF framework is equivalent to the energy guidance: TFTF samples from $p_{\text{data}}(x_1|y) \propto p_{\text{data}}(x_1)\ \tilde{p}(y|x_1)$. Since $\tilde{p}(y|x_1)$ can always be written as $e^{-J(x_1)}$ for some energy function $J$ in non-zero regions of $\tilde{p}(y|x_1)$, TFTF appears slightly more general as it permits $\tilde{p}(y|x_1) = 0$ for some $x_1$, whereas $e^{-J(x_1)} > 0$ always holds unless $J(x_1)$ is forced to be $+\infty$ in certain regions.
>
> ---
>
> ## 2. Relationship between Prop 3.1 and Thm 3.1 of [1]
>
> As discussed, we rewrite $e^{-J(x_1)}$ equivalently as $\tilde{p}(y|x_1)$ and set $P=1$ (also adopted in [1]). Thm 3.1 of [1] defines $Z_t(x_t) = \int P e^{-J(x_1)} p(z|x_t) dz$, where $z = (x_0, x_1)$, which automatically becomes $p(y|x_t)$. Further, $g_t(x_t) = \int \left[\frac{P e^{-J(x_1)}}{Z_t(x_t)} - 1\right] v_{t|z}(x_t|z) p(z|x_t) dz=\int \left[\frac{\tilde{p}(y|x_1)}{p(y|x_t)} - 1\right] v_{t|z}(x_t|z) p(z|x_t) dz$.
>
> Under our framework, $p(x_0)$ is the standard Gaussian and $v_{t|z}(x_t|z) = x_1 - x_0$. Using the identity $\nabla_{x_t} p(x_t|x_1) = -\frac{x_t - t x_1}{(1-t)^2} p(x_t|x_1)$ from our Appendix B.1, we can further derive the explicit form of $g_t(x_t)$ as:
> $$g_t(x_t)=\int \left[\frac{\tilde{p}(y|x_1)}{p(y|x_t)} - 1\right](x_1 - x_0) p(z|x_t) dz$$
> $$= \frac{1-t}{t}\int \left[\frac{\tilde{p}(y|x_1)}{p(y|x_t)} - 1\right] \left(-\frac{x_t - t x_1}{(1-t)^2}\right) p(x_t|x_1)\frac{p(x_1)}{p(x_t)} dx_1$$
> $$= \frac{1-t}{t}\int \left[\frac{\tilde{p}(y|x_1)}{p(y|x_t)} - 1\right] \nabla_{x_t} p(x_t|x_1) \frac{p(x_1)}{p(x_t)} dx_1$$
> $$= \frac{1-t}{t}\int \left[\frac{\tilde{p}(y|x_1)}{p(y|x_t)} - 1\right] \nabla_{x_t} p(x_1|x_t) dx_1$$
> $$= \frac{1-t}{t} \nabla_{x_t}\log p(y|x_t),$$
> which is precisely the guidance term decomposed in our Prop 3.1 (here we use the fact that $\int \left[\frac{\tilde{p}(y|x_1)}{p(y|x_t)} - 1\right] p(x_1|x_t) dx_1 = 0$).
>
> ---
>
> ## 3. Comparison and Discussion of [1], [2], [3]
>
> We now clarify the relationship of each reference to our work and the existing categories discussed in our Related Work (Section 5):
>
> **$g_t^{\text{MC}}$ [1] falls under the surrogate-based methods category discussed in our Section 5**. Specifically, $g_t^{\text{MC}}$ directly estimates the integral-form guidance term $g_t(x_t)$ via Monte Carlo sampling, which increases computational cost, may suffer from high variance, and is limited to low-dimensional settings (as acknowledged by [1] in their Section 3.2). In contrast, TFTF leverages the Gaussian assumption on the source distribution and linear interpolation structure  to transform the guidance term into $\frac{1-t}{t}\nabla_{x_t}\log p(y|x_t)$ and approximates it via a single-step Euler projection, which is computationally efficient. TFTF then corrects the resulting approximation error through reweighting and resampling, achieving **asymptotic accuracy that scales to high-dimensional settings** (as demonstrated in our experiments).
>
> **OC-Flow [2] falls under the optimization-based methods category discussed in our Section 5**, extending D-Flow (already compared in our paper). Unlike TFTF, OC-Flow is based on optimal control theory and cannot provide asymptotic accuracy guarantees for convergence to the conditional distribution. By contrast, it can **only bound the KL divergence from the unconditional data distribution (the prior)**. Moreover, generating a single sample requires multiple iterations of solving the entire ODE trajectory to optimize the control term.
>
> **FMPS [3]** is already included as a baseline in all our experiments (referred to as "FMPS" in our paper).
>
> We will add a more detailed discussion of [1] and [2] in the Related Work section of the final version.
>
> ---
>
> ## 4. Additional Quantitative Comparison
>
> To further substantiate our claims, we revisit the **Gaussian mixture example** (Section 4.1) and report the Sliced Wasserstein Distance (SWD) between each method's sampling distribution and the ground-truth target distribution (20,000 samples each):
>
> | Method | SWD ($\downarrow$) |
> |---|---|
> | TFTF (Ours) | **0.0316** |
> | $g_t^{\text{MC}}$ [1] | 0.1664 |
> | OC-Flow [2] | 0.4018 |
> | D-Flow | 0.4205 |
> | FlowChef | 0.5991 |
> | FMPS [3] | 1.2508 |
>
> **TFTF achieves an SWD an order of magnitude lower than the competing methods**, confirming that it converges to the target distribution while other methods exhibit remaining bias. This advantage stems directly from our principled importance weighting framework, which corrects for approximation errors rather than leaving them unaddressed.

---

> > ### Author Rebuttal · Reviewer_omqp · 2026-04-03
> >
> > Thank you for the response. I raise my score.

---

> > > ### Author Response · Authors · 2026-04-08
> > >
> > > We are grateful for the time and thoughtfulness you invested in reviewing our work. It is encouraging to know that our responses were able to resolve all of your concerns.

---

### Official Review · Reviewer_zt8k · 2026-03-11

**Soundness:** 3
**Presentation:** 3
**Significance:** 3
**Originality:** 4
**Overall Recommendation:** 4
**Confidence:** 4

**Summary:**

The paper proposes TFTF, a training-free conditional sampling framework for flow matching models that integrates importance sampling with sequential Monte Carlo (SMC) resampling. To enable particle resampling within the inherently deterministic flow-matching dynamics, the authors derive a family of stochastic flows (SDEs) whose time-marginals match the original ODE. Practically, they apply an elegant ODE–SDE–ODE sampling schedule, utilizing a one-step Euler projection to construct an intermediate look-ahead target that stabilizes importance weights. The method provides theoretical guarantees for asymptotic accuracy (under bounded weight conditions) and yields strong empirical results on a 2D toy problem, MNIST, CIFAR-10, and CelebA-HQ, demonstrating substantial improvements in effective sample size (ESS), external-classifier accuracy, and overall sample quality compared to existing baselines.

**Compliance With Llm Reviewing Policy:**

Affirmed.

**Final Justification:**

Thank you for the detailed rebuttal. My main concerns have been addressed, and I will maintain my score.

**Key Questions For Authors:**

1. Since TFTF outputs weighted particle sets, please clarify how evaluation metrics such as Inception Score, (W_2), and (ACC_E) are computed. In particular, is a final resampling step performed before evaluation?
2. Could the authors provide a stricter iso-compute comparison against baselines such as FlowChef or FMPS under matched sample or NFE budgets?
3. The theoretical exactness result is stated in continuous time, while the implementation relies on Euler-Maruyama discretization. It would be helpful to clarify how this discretization gap affects the practical validity of the asymptotic exactness claim.
4. The paper would be further strengthened by reporting more standard generative metrics (e.g., FID/sFID) and by expanding the discussion of closely related recent work on ODE-to-SDE conversions for flow models.

**Limitations:**

Not fully. While the paper includes an Impact Statement, it does not sufficiently discuss several important methodological limitations or the possibility of inherited bias from the external guidance models it relies on. In particular, TFTF incurs substantially higher inference-time compute and memory cost than single-particle training-free methods, and it also depends on heuristic design choices such as the resampling interval and the stochasticity schedule, as well as discretization of the continuous-time theory. In addition, because the method relies on external likelihood or guidance models such as classifiers or CLIP, it may inherit and potentially amplify biases present in those models. A brief “Limitations and Broader Impacts” paragraph acknowledging these points would strengthen the paper.

**Strengths And Weaknesses:**

**Strengths**
1. The paper introduces an elegant ODE\rightarrowSDE\rightarrowODE sampling schedule. Deriving the stochastic flow via the Fokker-Planck equation successfully enables Sequential Monte Carlo (SMC) resampling within inherently deterministic flow-matching dynamics.
2. The design of intermediate targets—using a one-step Euler projection as a look-ahead likelihood—is an innovative and practical workaround to guide particles early in the generation process.
3. The framework provides solid asymptotic correctness guarantees for sampling p_{data}(\cdot|y). Furthermore, the "extended target" mechanism smartly circumvents the need for expensive divergence computations during the final ODE segment.
4. The method successfully addresses a critical bottleneck in training-free conditional generation: the severe weight degeneracy problem of naive importance sampling in high-dimensional spaces.
5. The experimental validation is rigorous. Distinguishing between in-likelihood (ACC_L) and external-classifier (ACC_E) accuracies is a necessary and effective sanity check to prove the method does not merely adversarially exploit the guidance classifier.

**Weaknesses**
1. The theoretical exactness claim is derived in continuous time, whereas the practical algorithm relies on Euler-Maruyama discretization. The effect of this approximation error on the claimed asymptotic correctness is not fully discussed.
2. The intermediate target depends on a one-step Euler look-ahead projection, which is heuristic and may become inaccurate in more complex generation settings. The paper provides a limited analysis of this approximation.
3. Because TFTF outputs weighted sample sets, the evaluation protocol should be clarified more explicitly, especially for metrics such as Inception Score (W_2) and external-classifier accuracy.
4. The experimental comparison could be improved through stricter iso-compute analysis, and the paper would benefit from broader discussion of closely related ODE-to-SDE and SMC-based sampling methods.

---

> ### Author Rebuttal · Authors · 2026-03-30
>
> We sincerely thank the reviewer for providing us with many constructive comments. All concerns are addressed, and a point-by-point response is given below.
>
> ---
>
> ## 1. Analysis of the One-Step Euler Look-Ahead Projection
>
> We note that this step can also be viewed as **conditional expectation**. From Eq. (3) and Eq. (8), we obtain that $\hat{x}_{1}(x(t)) = \mathbb{E}[\tilde{X}(1) \mid \tilde{X}(t) = x(t)],$ which is the **MMSE-optimal point estimate** of the final state and thus the **best single-point imputation for the likelihood look-ahead**. Moreover, it **incurs no additional NFEs** beyond the velocity field evaluation already required at each step.
>
> ---
>
> ## 2. FID Evaluation
>
> We conduct FID measurements on CIFAR-10 (160K samples, 16K per class) and provide the pretrained unconditional model's FID as a reference lower bound:
>
> | Method | FID ($\downarrow$) |
> |---|---|
> | FMPS | 35.26 |
> | FlowChef | 18.99 |
> | TFTF ($K{=}16$; $M{=}1$) | 14.07 |
> | TFTF ($K{=}16$; $M{=}1000$) | 6.65 |
> | TFTF ($K{=}32$; $M{=}500$) | 6.79 |
> | Unconditional model (800 Euler steps) | 2.86 |
>
> Here $K$ and $M$ denote the per-node particle count and the number of nodes in Algorithm 2 (Appendix E). **TFTF outperforms all baselines and its FID improves as the total number of particles increases**, corroborating the asymptotic accuracy guarantee in Propositions 3.3 and E.1. As a training-free method, an FID of 6.65/6.79 demonstrates a strong distributional coverage.
>
> ---
> ## 3. Evaluation Protocol
>
> **ACC$_E$, ACC$_L$, CLIP$_L$** are computed as weighted averages. **IS, $W_2$, and FID** are computed by first resampling from the weighted particle set, then evaluating on the resulting equally weighted samples. We will explicitly document this in the final version.
>
> ---
>
> ## 4. Effect of Numerical Discretization
>
> This is an excellent question. We note that **all existing differential-equation-based generative models** incur this type of numerical errors from numerical solvers, whether for unconditional or conditional generation. We conjecture that DNN learning occurring at discretized points may alleviate the error accumulation.
>
> We show that the weighted particles converge to the extended target distribution (Eqs. (15)–(16)). Since the likelihood $\tilde{p}(y|\cdot)$ is evaluated exactly although the transition densities in Eq. (16) are computed via Euler–Maruyama discretization, we can claim that TFTF introduces no additional error beyond what is already present in the pretrained unconditional model.
>
> ---
>
> ## 5. Iso-Compute Comparison
>
> We have provided a computational complexity analysis with wall-clock time comparisons on CIFAR-10 in Table 5 (Appendix G.1). Due to the character limit of this rebuttal, please refer to **Part 3 of our response to Reviewer AesQ** for more details.
>
> ---
>
> ## 6. Related Work: ODE-SDE Conversions
>
> We appreciate this suggestion and discuss the most closely related works:
>
> - **Albergo et al. (2023)** provide a unifying framework for ODE-based and SDE-based models, deriving loss functions for learning velocity and score models. Having both networks enables the choice between ODE and SDE generation dynamics.
>
> - **Tong et al. (2024)** also require training two networks to support both dynamics.
>
> In contrast, our work focuses on **conditional generation within the flow matching framework**. By exploiting the mathematical structure of flow matching, our derived SDE requires **only the pretrained unconditional velocity field network**.
>
> **Refs:**
>
> Albergo et al. (2023). Stochastic interpolants: A unifying framework for flows and diffusions.
>
> Tong et al. (2024). Simulation-free Schrödinger bridges via score and flow matching.
>
> ---
>
> ## 7. Related Work: SMC-Based Sampling
>
> Prior works (Trippe et al., 2022; Cardoso et al., 2023; Wu et al., 2023) cited in Section 5 all apply SMC to **stochastic diffusion models**. TFTF addresses a fundamentally different setting: flow matching with **inherently deterministic** dynamics, where we inject stochasticity through our derived stochastic flow. To our knowledge, TFTF is the **first** to apply SMC to flow matching via an ODE→SDE→ODE strategy for asymptotically exact conditional generation.
>
> Furthermore, **compared to prior diffusion-based SMC methods that simulate SDE with resampling throughout, our ODE→SDE→ODE strategy offers a distinct advantage: resampling is most effective during intermediate stages**. Too early, and samples are too noisy for reliable discrimination; too late, and coarse-scale structure is already determined, so resampling merely eliminates diversity. **Stochasticity is therefore only useful around the resampling steps**, where it enables particles to branch out. In the early and late stages, SDE's added noise becomes redundant and ODE is more efficient in these regimes. Empirically, Figure 4 shows that TFTF achieves **superior diversity and classification accuracy** compared to TDS (Wu et al., 2023). We will expand this discussion in the final version.

---

> > ### Author Rebuttal · Reviewer_zt8k · 2026-04-03
> >
> > All my concerns have been addressed.

---

> > > ### Author Response · Authors · 2026-04-05
> > >
> > > We are grateful for the time and thoughtfulness you invested in reviewing our work. It is encouraging to know that our responses were able to resolve all of your concerns.

---

### Official Review · Reviewer_HQy9 · 2026-03-11

**Soundness:** 3
**Presentation:** 3
**Significance:** 2
**Originality:** 2
**Overall Recommendation:** 4
**Confidence:** 3

**Summary:**

The paper proposes TFTF, a training-free approach for conditional generation.
The paper assume access to a pretrained unconditional flow matching model in addition to a conditional density estimator. The authors address a limitation of traditional importance sampling (IS),weight degeneracy in high-dimensional spaces, by integrating Sequential Monte Carlo (SMC) resampling at intermediate stages of the generation process.
To enable effective resampling within the deterministic dynamics of flow matching, the authors derive a stochastic flow.

The method is theoretically grounded, providing guarantees of asymptotic accuracy for sampling from the target conditional distribution.
The authors evaluate TFTF against existing training-free baselines in the tasks of class-conditional sampling on MNIST and CIFAR-10, and text-to-image generation on CelebA-HQ.

**Compliance With Llm Reviewing Policy:**

Affirmed.

**Final Justification:**

The rebuttal solved my concerns.

**Key Questions For Authors:**

* Given the requirement of $K=25$ particles, what is the peak GPU memory usage when scaling TFTF to larger high-resolution models, and is there a strategy to reduce this overhead for consumer-grade hardware?
* Did the authors test their method on large scale unconditional image generation models (e.g. FLUX with null prompt)? Since the approach is training-free, it is interesting to see its performance on real-world scenarios. Qualitative comparison would suffice.

**Limitations:**

yes

**Strengths And Weaknesses:**

Strengths:

* The method significantly outperforms optimization-based (D-Flow) and surrogate-based (FMPS, FlowChef) methods in classification accuracy.
* This work provides a rigorous proof of asymptotic accuracy. It demonstrates that as the number of particles $K$ increases, the samples converge to the true target distribution.

Weaknesses:

* The authors motivate the need for a training-free method by stating that training-based approaches depend on labeled data, which can be "scarce and costly to acquire". However, the proposed method requires an "off-the-shelf" likelihood estimator. Since training these models requires the same labeled data that the authors claim is difficult to obtain, the practical advantage of this method over training-based conditional models is not clearly justified.
* In the CelebA-HQ experiments, the likelihood is computed using a specific positive text prompt against multiple negative prompts. This setup suggests the method might not be fully "open-vocabulary" and could struggle if appropriate negative constraints are not provided.
* The evaluation lacks Fréchet Inception Distance (FID), which is the standard metric in generative modeling. This is critical in order to assess the realism and diversity of the generated images.

---

> ### Author Rebuttal · Authors · 2026-03-30
>
> We sincerely thank the reviewer for providing us with many constructive comments. All concerns are addressed, and a point-by-point response is given below.
>
> ---
>
> ## 1. Practical Advantage of the Training-Free Paradigm
>
> We appreciate this important question, which gives us an opportunity to improve our writing.
>
> Our method assumes access to (1) a pretrained unconditional model and (2) a likelihood function, which can be either **theory-based** (e.g., molecular force fields in protein design, where domain knowledge specifies the likelihood without any labeled data) or **prior-data-based** (e.g., a CLIP model learned from labeled data from other sources). In both cases, the TF-paradigm has the following advantages, which will be stated more clearly in the revision:
>
> - **No need of a direct access to labeled data.** We only need access to the likelihood model itself, not the data it was trained on.
>
> - **The choice of the likelihood model is entirely independent of the generative model framework.** One can directly plug in any publicly available, high-performing models that operate on **noise-free** samples.  In contrast, **for training-based methods,  one must have direct access to labeled data to retrain or fine-tune the generative model**, which is computationally expensive.
>
> - **Training-free is particularly useful in embodied-AI systems** when a quick reaction and model update are needed.
>
> ---
>
> ## 2. CelebA-HQ Likelihood Construction
>
> We thank the reviewer for raising this point. The positive-vs-negative-prompt softmax likelihood used is just **one particular instantiation**. In general, for any given likelihood, TFTF samples from the corresponding target distribution $p_{\text{data}}(\cdot|y) \propto p_{\text{data}}(\cdot)\\tilde{p}(y|\cdot)$; hence, the quality of the generated samples depends on how well the chosen likelihood captures the desired condition. We note that **how to construct the likelihood** and **how to sample from the resulting conditional distribution** are two orthogonal problems, and TFTF addresses the latter. To demonstrate, we provide supplementary results for **an alternative likelihood that does not require negative prompts**. Specifically, we define the likelihood as the exponential of the cosine similarity between the image features and the positive prompt:
>
> | Method | CLIP$_L$ | ACC$_E$ ($\uparrow$) | ESS |
> |---|---|---|---|
> | FMPS | 0.284 | 60.19% | – |
> | FlowChef | 0.271 | 29.86% | – |
> | TFTF (Ours) | 0.258 | 73.25% | 22.70/25 |
>
> Under this new likelihood, all methods show some performance degradation, but **TFTF still performs the best**. We acknowledge that for simple tasks, the likelihood function is straightforward to define, whereas more complex tasks may require more careful likelihood design. We will include a thorough discussion of likelihood selections in the final version.
>
> ---
>
> ## 3. FID Evaluation
>
> We thank the reviewer for this suggestion. We have computed FID scores for the CIFAR-10 class-conditional generation experiments. We also report the FID of the pretrained unconditional model as a lower-bound reference:
>
> | Method | FID ($\downarrow$) |
> |---|---|
> | FMPS | 35.26 |
> | FlowChef | 18.99 |
> | TFTF ($K{=}16$; $M{=}1$) | 14.07 |
> | TFTF ($K{=}16$; $M{=}1000$) | 6.65 |
> | TFTF ($K{=}32$; $M{=}500$) | 6.79 |
> | Unconditional model (800 Euler steps) | 2.86 |
>
> Each method generates a total of 160K samples (16K per class). Here $K$ and $M$ denote the per-node particle count and the number of nodes in Algorithm 2 (Appendix E). **TFTF  outperforms all baselines and its FID improves as the total number of particles increases**, which corroborates the asymptotic accuracy guarantee established in Proposition 3.3 and Proposition E.1. As a training-free method, an FID of 6.65/6.79 demonstrates a wide distributional coverage.
>
> ---
>
> ## 4. GPU Memory on High-Resolution Data
>
> As described in Section 4.3, for CelebA-HQ ($256 \times 256 \times 3$), we employ the **accelerated variant** of TFTF. This variant skips backpropagation through the UNet, which **reduces both computation time and GPU memory consumption**. Concretely, on an RTX 4090 with $K{=}25$ particles, the **peak GPU memory usage is 20,867 MiB out of 24,564 MiB available** and the ability to run on such hardware is precisely enabled by the accelerated variant (Appendix G.1). We will clarify this point in the final version.
>
> ---
>
> ## 5. Experiments on FLUX
>
> Following the reviewer's suggestion, we have conducted **style-guided generation experiments on FLUX**. Given a reference style image, we define the likelihood as the exponential of the Gram matrix similarity between the reference image and the generated image.  Qualitative results are available [**here (PDF)**](https://anonymous.4open.science/r/2026-814E), which demonstrate that TFTF can be  applied to real-world models in a plug-and-play manner, further validating the practical applicability of our TF paradigm.

---

> > ### Author Rebuttal · Reviewer_HQy9 · 2026-04-03
> >
> > The rebuttal solved my concerns and I decided to raise my score to 4.

---

> > > ### Author Response · Authors · 2026-04-08
> > >
> > > We are grateful for the time and thoughtfulness you invested in reviewing our work. It is encouraging to know that our responses were able to resolve all of your concerns.

---

### Decision · Program_Chairs · 2026-04-30

**Decision:**

Accept (regular)

**Comment:**

Four knowledgeable reviewers went over this submission. The main concerns raised by the reviewers were:

1. Unconvincing experimental evidence (omqp, HQy9, zt8k):
    - Missing comparisons positioning and comparisons with recent methods including energy guidance, OC-Flow, FMPS.
    - Missing standard metrics (e.g., FID)
2. Unclear practical justification for the proposed approach (HQy9)
3. Gap between continuous-time theory and discretization in practice, stability around t=0 (zt8k, AesQ)
4. Scalability and/or compute overhead (HQy9, zt8k, AesQ)
5. Presentation could be improved by reducing the amount of notation (AesQ)

During rebuttal, the authors clarified the positioning of the work w.r.t. each suggested method, added quantitative comparisons highlighting the benefits of TFTF, included FID results on CIFAR and wall-clock comparisons showing competitive performance. The rebuttal also emphasized that the key advantage is modularity, and argued that TFTF introduces no additional discretization error beyond the one of the pre-trained model. All four reviewers considered their concerns fully addressed post-rebuttal. The paper's contribution was recognized as novel and sound, and the rebuttal strengthened the empirical validation, making it convincing. All reviewers reached consensus to accept the contribution. The AC agrees with their assessment and recommends to accept.